# $N^6$-methyladenosine modification is not a general trait of viral RNA genomes

Belinda Baquero-Pérez [1,6], Ivaylo D. Yonchev [2,3,6], Anna Delgado-Tejedor [4,6], Rebeca Medina[4], Mireia Puig-Torrents[1], Ian Sudbery [2], Oguzhan Begik [4], Stuart A. Wilson [2] ✉, Eva Maria Novoa [4,5] ✉ & Juana Díez [1] ✉

Despite the nuclear localization of the m⁶A machinery, the genomes of multiple exclusively-cytoplasmic RNA viruses, such as chikungunya (CHIKV) and dengue (DENV), are reported to be extensively m⁶A-modified. However, these findings are mostly based on m⁶A-Seq, an antibody-dependent technique with a high rate of false positives. Here, we address the presence of m⁶A in CHIKV and DENV RNAs. For this, we combine m⁶A-Seq and the antibody-independent SELECT and nanopore direct RNA sequencing techniques with functional, molecular, and mutagenesis studies. Following this comprehensive analysis, we find no evidence of m⁶A modification in CHIKV or DENV transcripts. Furthermore, depletion of key components of the host m⁶A machinery does not affect CHIKV or DENV infection. Moreover, CHIKV or DENV infection has no effect on the m⁶A machinery's localization. Our results challenge the prevailing notion that m⁶A modification is a general feature of cytoplasmic RNA viruses and underscore the importance of validating RNA modifications with orthogonal approaches.

$N^6$-methyladenosine (m⁶A) is the most abundant internal modification on eukaryotic mRNAs, present at a frequency of ~0.15–0.6% of all adenosines[1]. m⁶A regulates major aspects of RNA metabolism, including RNA stability[2,3], cap-dependent and independent translation[4,5], nuclear export[6,7] and splicing[8]. m⁶A decoration of mRNAs is a dynamic process that requires m⁶A-methyltransferases (writers) to add m⁶A modifications, m⁶A-demethylases (erasers) to remove them, and m⁶A-binding proteins (readers) to bind to them. m⁶A addition occurs co-transcriptionally in the nucleus[9,10] by the methyltransferase writer complex (MTC), composed of the methyltransferase-like 3 (METTL3) and accessory proteins such as methyltransferase-like 14 (METTL14) and Wilm's-tumor-1-associated protein (WTAP)[11–13]. The MTC adds m⁶A at a consensus sequence whose core is typically represented by the DRACH motif (D = A, G or U; R = G or A; H = A, C or U)[14]. Noteworthy, not all DRACH motifs are modified and around 6% of

METTL3-mediated m⁶A modifications occur outside the DRACH motif[15]. The erasers fat-mass- and obesity-associated protein (FTO)[16] and AlkB homolog 5 (ALKBH5)[17] remove m⁶A modifications. Finally, the readers that recognize and bind m⁶A sites include proteins from the YTH domain-containing family proteins (YTHDF1-3 and YTHDC1-2)[18]. All components of the writing and erasing m⁶A machinery have a marked nuclear localization[11,13,19–21]. By contrast, readers have a predominantly cytoplasmic localization[2,4,22–24], except for YTHDC1, which localizes in the nucleus[8].

The interest in dissecting the location and function of m⁶A modifications in host mRNAs and viral RNAs was re-ignited following the development of transcriptome-wide RNA modification mapping methods. These methods, in contrast to liquid chromatography with tandem mass spectrometry (LC–MS/MS), allow precise identification of m⁶A locations. Moreover, they avoid potential misinterpretations of

[1]Molecular Virology Group, Department of Medicine and Life Sciences, Universitat Pompeu Fabra, Dr. Aiguader 88, 08003 Barcelona, Spain. [2]Sheffield Institute for Nucleic Acids (SInFoNiA) and School of Biosciences, The University of Sheffield, Firth Court, Western Bank, Sheffield S10 2TN, UK. [3]Department of Molecular and Cellular Biology, Baylor College of Medicine, Houston, TX 77030, USA. [4]Center for Genomic Regulation (CRG), The Barcelona Institute of Science and Technology, Dr. Aiguader 88, 08003 Barcelona, Spain. [5]Universitat Pompeu Fabra (UPF), 08003 Barcelona, Spain. [6]These authors contributed equally: Belinda Baquero-Pérez, Ivaylo D. Yonchev, Anna Delgado-Tejedor. ✉e-mail: stuart.wilson@sheffield.ac.uk; eva.novoa@crg.eu; juana.diez@upf.edu

the results caused by the presence of cellular RNAs that even in trace amounts might affect m⁶A quantifications of viral RNAs when using LC-MS/MS. Many m⁶A mapping methods rely on fragmentation and immunoprecipitation of m⁶A sites (m⁶A-IP) by specific anti-m⁶A antibodies. These antibody-dependent methods are coupled to either qRT-PCR analysis, in which the presence of m⁶A is interrogated in RNA fragments that vary between 100 to 1000 nucleotides (nt) (m⁶A-IP-qRT-PCR), or to high-throughput sequencing. Among the latter, m⁶A-Seq[25,26] is one of the most widely used next-generation sequencing (NGS)-based methods to detect m⁶A modifications transcriptome-wide. m⁶A-Seq identifies regions of m⁶A enrichment (m⁶A peaks) in the distribution of reads from an m⁶A-IP sample when compared to a non-immunoprecipitated input control, with a resolution of ~200 nt. More laborious methods that add UV-crosslinking, such as photo-crosslinking-assisted m⁶A-sequencing (PA-m⁶A-Seq)[27] and m⁶A individual-nucleotide resolution UV crosslinking and immunoprecipitation (miCLIP)[28] permit detection of m⁶A modifications at higher resolution. However, antibody-dependent m⁶A detection methods have high false positive rates and low reproducibility[29,30], as similarly shown for antibody-dependent detection methods against other RNA modifications[31–33]. More recently, novel antibody-independent methods have been developed that allow m⁶A mapping at single-nucleotide resolution, such as SELECT, which uses a single-base elongation and ligation-based qRT-PCR amplification to identify m⁶A modifications at specific nucleotide locations[34]. In addition, nanopore direct RNA sequencing (DRS), which is an antibody-independent method that allows sequencing of the native RNA molecules without reverse transcription or PCR amplification, can now provide transcriptome-wide maps of m⁶A modifications in native full-length RNA sequences[35–37].

In recent years, several works using LC−MS/MS and/or m⁶A-Seq— or other antibody-based techniques—have reported internal m⁶A modifications in viral RNA transcripts from both nuclear- and cytoplasmic-replicating viruses[38]. The latter include positive-strand (+) RNA viruses such as hepatitis C virus (HCV)[39], dengue virus (DENV)[39], zika virus (ZIKV)[39,40], and chikungunya virus (CHIKV)[41] which were all shown to be abundantly m⁶A-modified. Yet, how these cytoplasmic viral RNAs are modified by the nuclear MTC, a large protein complex of ~1000 kDa in size[42], and what the biological relevance of such modifications is remains unclear. To address these questions, here we use CHIKV and DENV, two viruses that replicate to high levels, as model systems, which can facilitate m⁶A detection. More specifically, we comprehensively analyze the presence of m⁶A modifications in CHIKV and DENV RNAs by intersecting antibody-dependent (m⁶A-Seq) and antibody-independent (SELECT and nanopore DRS) techniques in different cell lines. Additionally, we incorporate the use of in vitro transcribed (IVT) viral RNAs as negative controls and perform site-directed mutagenesis as well as functional studies. To our surprise, we find no evidence of m⁶A modifications in either virus, challenging the idea that cytoplasmic-replicating RNA viruses are abundantly modified and highlighting the fundamental importance of orthogonal validation of RNA modification detection.

## Results

### Mass spectrometry and m⁶A-IP-qRT-PCR analyses of CHIKV RNA do not detect m⁶A enrichment

The CHIKV (+) RNA genome (gRNA) consists of a 5′ capped and 3′ poly(A)-tailed single-stranded RNA that contains two open reading frames (ORFs). The first ORF encodes four non-structural proteins required for RNA replication. The second ORF is expressed from a 5′ capped and 3′ poly(A)-tailed subgenomic RNA (sgRNA) transcribed during infection and encodes five structural proteins found within the virion (Fig. 1a). To address whether CHIKV RNAs are enriched in m⁶A modification, we first performed liquid chromatography with tandem mass spectrometry (LC-MS/MS). Poly(A) + RNA was isolated from mock- or CHIKV-infected HEK293T and Huh7 cells at 12 h post-

infection (p.i.), a time at which ~50% of ribodepleted RNA reads correspond to CHIKV RNAs[43]. Surprisingly, RNA samples isolated from both cell lines exhibited no enrichment in m⁶A modification following viral infection (Fig. 1b), instead CHIKV-infected Huh7 cells showed a significant decrease in m⁶A modification. These data suggested that m⁶A modification may not be abundant or may even be absent in CHIKV RNAs. A previous study identified m⁶A modification within the first two kilobases of the CHIKV RNA genome by m⁶A-IP-qRT-PCR[41]. Thus, as a first step to map m⁶A modification in CHIKV RNAs, we performed parallel m⁶A-IP-qRT-PCR from CHIKV-infected cells in three different cell lines. Total RNA was fragmented to ~1 kb-long fragments, immunoprecipitated with m⁶A antibody, and following reverse transcription, primers (Supplementary Data 1) were tiled along the CHIKV genome every ~1 kb, as previously described[41] (Fig. 1c). As positive and negative controls of m⁶A enrichment, we used primers (Supplementary Data 1) spanning a known m⁶A-modified[25] or non-modified region of the cellular *SLC39A14* transcript, respectively (Fig. 1d). As expected, significant m⁶A enrichment was found over the known m⁶A peak within the cellular *SLC39A14* transcript (Fig. 1e). However, no enrichment was detected for any of the viral regions. Less than ~1% of input was recovered for all viral regions (see Source Data Fig. 1e), a percentage lower than that recovered for the non-modified *SLC39A14* region, even when the input CHIKV RNA level in total RNA was ~10⁴- and ~4 × 10⁴-fold higher than that of the *SLC39A14* transcript in HEK293T- and U2OS-infected cells, respectively (Fig. 1f).

### m⁶A-Seq analysis of the CHIKV RNA genome reveals a single m⁶A peak

To confirm these results, we performed m⁶A-Seq with poly(A)+ selected RNAs samples (fragmented to ~100−200 nt) from CHIKV-infected HEK293T-infected cells (12 h p.i.). To evaluate the quality of our m⁶A-Seq dataset, we first performed peak-calling on cellular RNA using m6aViewer, a commonly used software that identifies high-confidence methylated adenosines[44]. Applying a cut-off of >2-fold enrichment of m⁶A-IP over input reads and a false discovery rate (FDR) < 5% across two biological replicates, peak-calling detected 23,539 m⁶A peaks (Supplementary Data 2). DRACH motif was identified as the most significantly enriched by HOMER motif analysis[45] (Fig. 2a). In CHIKV RNA, a single peak with 2.8-fold enrichment over input (Fig. 2b) was the only significant peak detected by two widely used m⁶A peak-calling programs, m6aViewer and MACS2[46] (Supplementary Data 2). In contrast, the known m⁶A site within the *SLC39A14* cellular transcript fell within the top 5% most highly enriched peaks with a 52-fold enrichment (Fig. 2c, d), despite having much lower abundance than CHIKV RNA (50% of all input reads were viral while <0.01% mapped to *SLC39A14*, Supplementary Table 1). The 2.8-fold enrichment seen at the viral peak was surprisingly low when compared with the 10.3-fold median m⁶A peak enrichment of the cellular dataset and fell within the 5% most lowly enriched peaks (Fig. 2d). Moreover, this viral peak could not be validated through m⁶A-IP-qRT-PCR using total RNA from CHIKV-infected cells, which was fragmented to a size comparable to that used in the m⁶A-Seq experiments (Fig. 2e). Together, these results suggested that the detected CHIKV m⁶A peak was either a false positive peak or that the m⁶A stoichiometry in this region is extremely low.

### SELECT and nanopore direct RNA sequencing (DRS) do not detect m⁶A modifications in CHIKV RNA

To validate our findings by antibody-free orthogonal methods, we used both SELECT and nanopore DRS. The SELECT technique is based on the m⁶A's ability to hinder the single-base elongation of *Bst* 2.0 DNA polymerase and the nick ligation of splintR ligase[34]. If the A in the DRACH motif is modified, the number of PCR cycles should be reduced when compared to a proximal non-modified adenosine (Fig. 3a). As the annealing efficiencies of the oligos in both locations might differ, it is important to confirm that such reduction is not observed in parallel

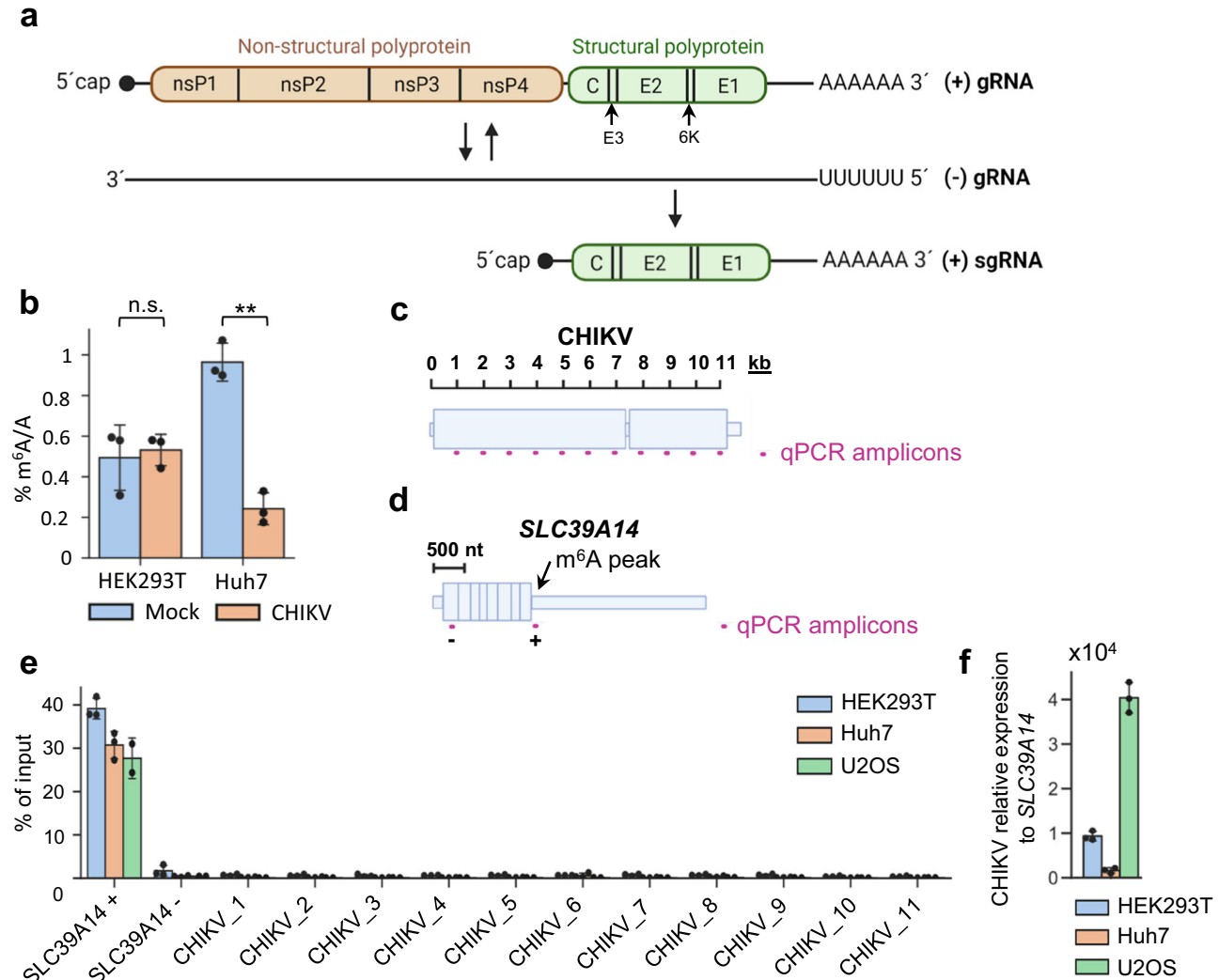

**Fig. 1 | Mass spectrometry and m⁶A-IP-qRT-PCR analyses of CHIKV RNA reveal no m⁶A modification.** **a** Schematic diagram of the genomic RNA (gRNA) and subgenomic RNA (sgRNA) generated during CHIKV infection. **b** LC-MS/MS quantification of m⁶A modification in poly(A) + RNA isolated from mock- or CHIKV-infected HEK293T and Huh7 cells. Cells were infected for 12 h, at a multiplicity of infection (MOI) of 4. -100 ng of digested ribonucleosides were analyzed for HEK293T samples and -75 ng per Huh7 samples. The bar chart shows mean values from three biological replicates with the error bars showing standard deviation (SD). n.s. not significant. **p < 0.01, using an unpaired two-tailed t-test, p value = 0.0005. Source data are provided with this paper. **c** Depiction of the qPCR amplicons tiled along the CHIKV RNA for m⁶A-IP-qRT-PCR analysis. **d** Depiction of the negative (−) and positive (+) control qPCR amplicons in the host SLC39A14 transcript for m⁶A-IP-qRT-PCR analysis. **e** m⁶A-IP-qRT-PCRs performed with total RNA isolated from different CHIKV-infected cell lines. The RNA was fragmented to -1 kb. 11 primer sets (one for every kb: CHIKV_1 to CHIKV_11)[41] were tiled along the CHIKV RNA. Bars represent the mean ± SD values of 3 m⁶A-IPs from 3 independent infections for HEK293T and Huh7 cell lines, and from 2 independent m⁶A-IPs for the U2OS cell line. All cell lines were infected for 12 h at an MOI of 4. Source data are provided with this paper. **f** Intracellular SLC39A14 and viral RNA levels were quantified in total RNA samples by qRT-PCR and normalized against the housekeeping gene GAPDH. For each cell line, SLC39A14 RNA levels were set to 1 and viral RNA levels expressed relative to those of SLC39A14. The bar chart shows mean values of 3 independent infections with the error bars showing SD. All cell lines were infected for 12 h at an MOI of 4. Source data are provided with this paper.

analyses with an in vitro transcribed (IVT) control, which is free of modifications. With SELECT, we addressed putative m⁶A modifications within the eight DRACH motifs identified within the 300 nt-long CHIKV m⁶A peak (8614–8914 nt) detected by m⁶A-Seq. As a positive control, an m⁶A-modified site in the SLC39A14 mRNA (genomic coordinate chr8:22419678) was used. This site was selected by identifying overlapping signals from public PA-m⁶A-IPs[27] and our m⁶A-IPs (Supplementary Fig. 1a). For the SLC39A14 mRNA motif, a 5.14 threshold cycle difference of amplification ($\Delta C_T$) was observed between the PCR products generated from the DRACH motif and the control oligos (Fig. 3b and Supplementary Fig. 1b), indicating the presence of m⁶A modification. Moreover, no difference in amplification cycles was seen using the in vitro control ($\Delta C_T$ of 0.65), confirming that the motif is modified. In contrast, the eight putative motifs in the CHIKV m⁶A peak

showed no significant reduction in the number of PCR cycles between the total RNA $\Delta C_T$ and the IVT $\Delta C_T$ (Fig. 3b and Supplementary Fig. 1b).

To further validate our results, we examined the RNA modification landscape of CHIKV RNA using the nanopore direct RNA sequencing (DRS) platform from Oxford Nanopore Technologies (ONT). This platform permits the sequencing of full-length native RNA molecules, thus retaining RNA modification information from each read. Once the RNA molecules have been sequenced, RNA modifications can be detected in the form of systematic base-calling "errors" and/or in the form of alterations in the current intensity[35,47]. To examine the presence of m⁶A RNA modifications in the CHIKV genomic and subgenomic RNAs, poly(A)+ selected RNA samples from CHIKV-infected HEK293T cells (12 h p.i.) were sequenced using DRS, in biological duplicates (see "Methods") (Supplementary Table 2). CHIKV IVT RNA

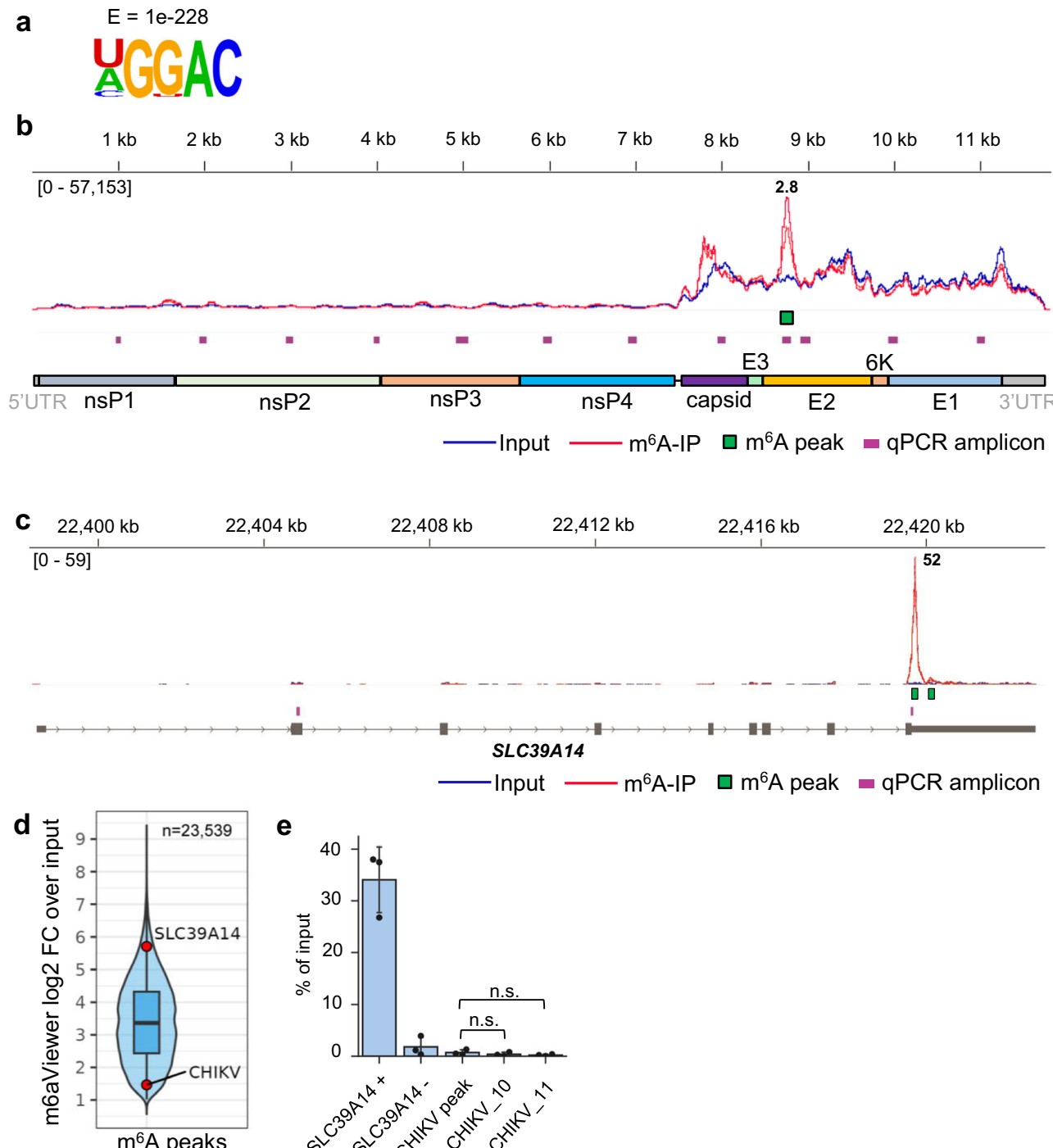

**Fig. 2 | m⁶A-Seq analysis of CHIKV RNA reveals a single putative m⁶A peak.**
**a** Most significantly enriched motif identified in conserved m⁶A peaks across cellular poly(A) + RNA in CHIKV-infected cells. HOMER software was used for motif analysis. **b** Genome browser tracks showing mapped CHIKV reads (strain LR2006-OPY1) for input and m⁶A-IP samples. The reads are scaled to the same read depth against the viral genome using counts per million normalization. Both biological replicates for each condition (input and m⁶A-IP) are displayed within each track. The common m⁶A peak identified by m6aViewer and MACS2 is indicated. The fold change of m⁶A-IP/input, averaged from two replicates is shown above the called m⁶A peak. HEK293T cells were infected for 12 h at an MOI of 4. The 11 qPCR amplicons generated by the 11 primer sets (CHIKV_1 to CHIKV_11) are indicated. The qPCR amplicon spanning the CHIKV m⁶A peak is also shown. **c** Genome browser tracks showing mapped reads for input and m⁶A-IP samples across the *SLC39A14* transcript. The reads are scaled against the merged genome using counts per million (CPM) normalization. The *SLC39A14* qPCR amplicons used as negative and positive controls in m⁶A-IP-qRT-PCRs are also indicated. **d** Violin and box plot of the log2 fold-change (log2FC) distribution of all 23,539 m6aViewer-called cellular peaks in CHIKV-infected HEK293T cells, conserved between two independent replicates. The *SLC39A14* and CHIKV peaks are indicated by red dots. The boxplot shows the median (middle line), 25–75th percentile values and 1.5x interquartile range, while the overlaid violin plot shows the full data distribution. **e** m⁶A-IP-qRT-PCRs performed to validate the putative m⁶A peak identified via m⁶A-Seq. HEK293T cells were infected for 12 h at an MOI of 4. Total RNA was fragmented to 100–200 nt. The bar chart shows mean values from 3 m⁶A-IPs from 3 independent infections with the error bars showing SD. The CHIKV_10 and CHIKV_11 primer sets were used as negative controls amplifying regions without m⁶A enrichment, according to our m⁶A-Seq data. n.s. not significant ($p > 0.05$, using the two-tailed $t$-test). Source data are provided with this paper.

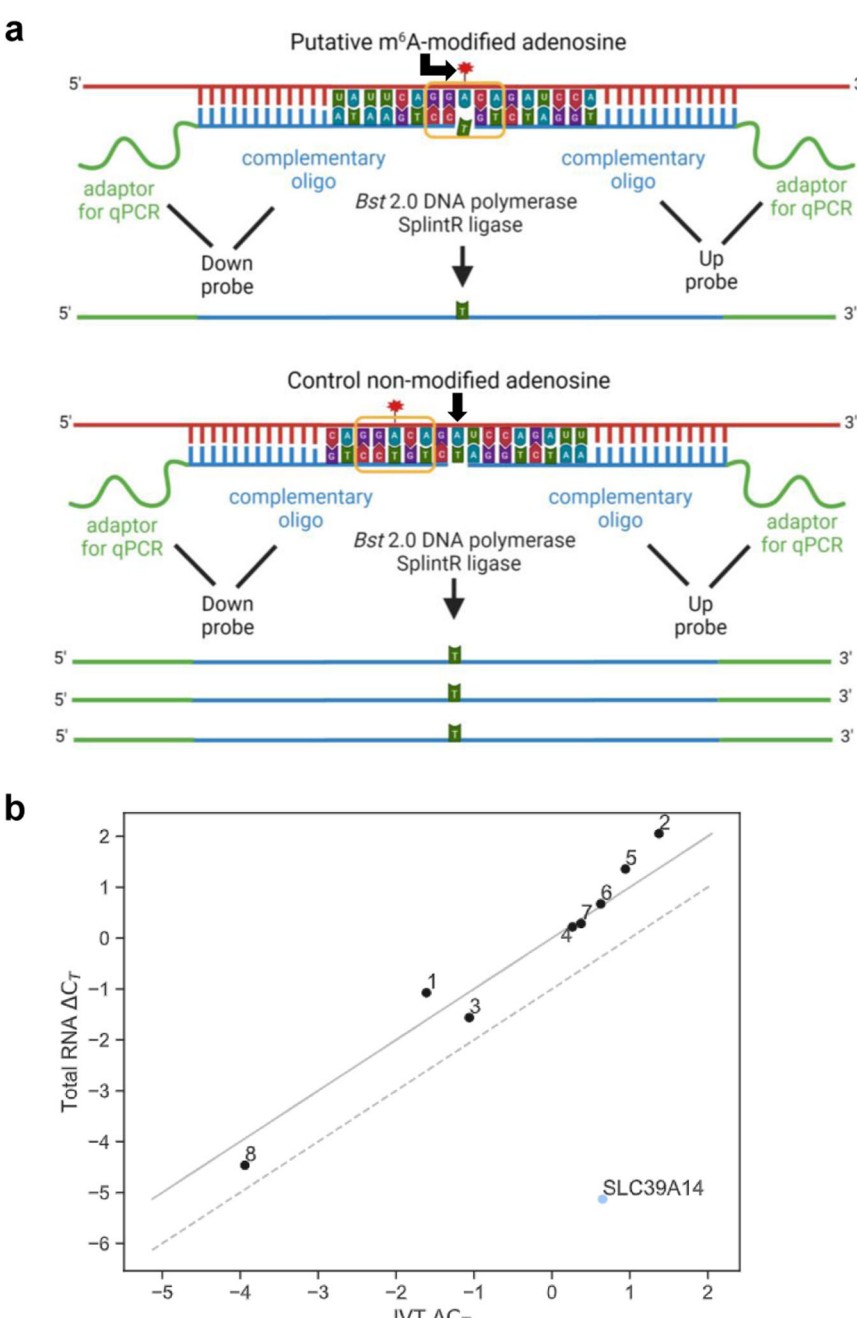

**Fig. 3 | SELECT analysis of CHIKV RNA reveals no m⁶A modification. a** Schematic diagram of the SELECT technique[34]. We designed DNA oligos that either annealed to the putative modified DRACH (leaving a gap at the modified site) or to an adjacent non-modified adenosine to serve as a control. The abundance of elongated and ligated products is reduced in the presence of m⁶A modification. The DRACH motif is highlighted with a yellow line. **b** SELECT technique performed with either total RNA or CHIKV in vitro transcribed (IVT) RNA. HEK293T cells were infected for 12 h at an MOI of 4. The threshold cycle difference of amplification ($\Delta C_T$) of total RNA versus IVT $\Delta C_T$ is shown for each motif (1–8). If total RNA $\Delta C_T$ = IVT $\Delta C_T$, values would align on the solid diagonal line. If there is a difference of −1 $C_T$ cycle between the total RNA $\Delta C_T$ and the IVT $\Delta C_T$, which would indicate the motif is m⁶A-modified, values would align on the diagonal discontinuous line. The lower this $C_T$ cycle difference between the total RNA $\Delta C_T$ and the IVT $\Delta C_T$, the more m⁶A modification is present in the DRACH motif. All experiments were performed in three technical replicates (separate SELECT reactions). All $C_T$ values from SELECT results can be found in Supplementary Fig. 1b and Source Data. Results are representative of two independent infections.

was also sequenced as a control. Detection of RNA modification in DRS data was performed by pairwise comparison of CHIKV-infected and CHIKV IVT RNA, to identify potential differentially modified sites between the two samples, as previously described[48]. More specifically, we used the software *NanoConsensus*[49], which uses predictions from diverse RNA modification software as input (*EpiNano*, *Nanocompore*, *Nanopolish* and *Tombo*), to identify possible RNA modified sites in the CHIKV RNA. Comparison of reads aligning to the CHIKV genome from CHIKV-infected HEK293T RNA to those obtained from CHIKV IVT RNA revealed no replicable differentially modified sites between the two samples (Fig. 4a and see also Supplementary Table 3), suggesting that the CHIKV genome is not m⁶A-modified, in agreement with our previous observations.

To confirm that the reason for these negative results were not the result of our data processing and analysis through the *NanoConsensus* pipeline, we examined whether it could identify previously reported

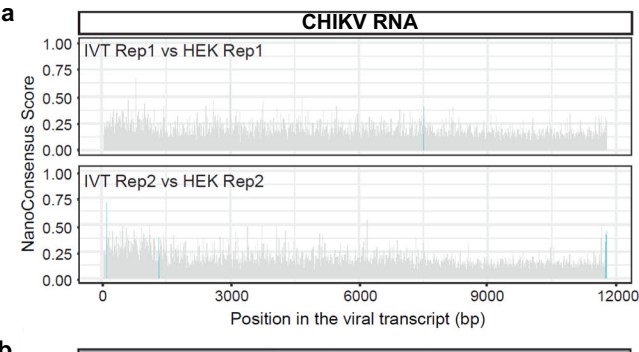

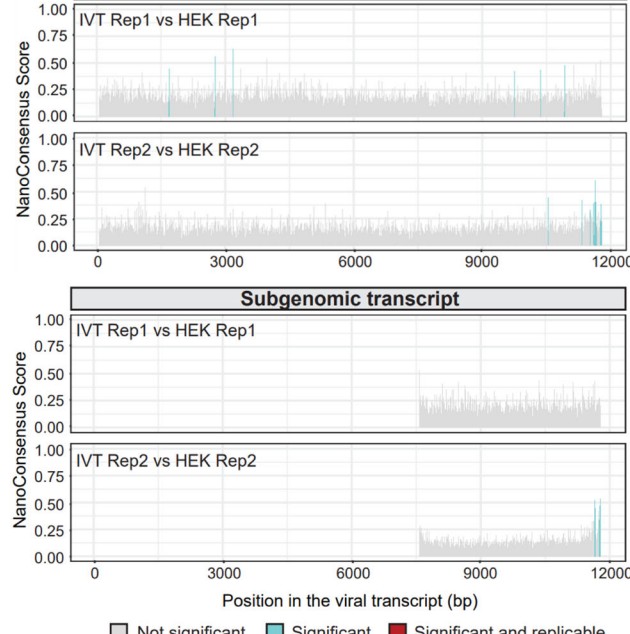

**Fig. 4 | DRS analysis of CHIKV RNA does not reveal modification within the antibody-dependent m⁶A peak. a** *NanoConsensus* scores along the CHIKV genome in HEK293T infected samples (12 h p.i, MOI of 4) obtained when compared to a CHIKV IVT sample control (using default parameters). In gray, non-significant positions; in blue, regions identified by *NanoConsensus* in only one replicate; in red, regions identified in both replicates. **b** *NanoConsensus* scores along the genomic (upper panels) and subgenomic transcript (lower panels) of CHIKV (using default parameters). In gray, non-significant positions; in blue, regions identified by *NanoConsensus* in only one replicate; in red, regions identified in both replicates. We should note that the potential modifications identified in one of the replicates at the very end of the transcript are typically false positives, caused by the high noise of nanopore signal at the ends of reads.

m⁶A-modified RNA sites in the host (human) transcriptome. To this end, we compared the nanopore reads from CHIKV-infected HEK293T cells mapping to human transcripts to those reads obtained from sequencing the IVT human transcriptome[50]. Our approach identified multiple replicable m⁶A modified sites (i.e., differentially modified regions between CHIKV-infected HEK293T and human IVT) in the host mRNA transcripts across a wide variety of transcripts (Supplementary Fig. 2 and see also Supplementary Table 4). This confirmed that our pipeline can accurately identify m⁶A sites via pairwise comparison of native RNA reads and IVT reads. Moreover, several of the identified m⁶A sites overlapped with previously reported m⁶A sites in HEK293 cells identified using miCLIP[28], confirming the accuracy of our pipeline.

We next examined whether the difference in coverage between the genomic and subgenomic region could be hijacking the performance of the algorithm. In other words, we wondered whether

subgenomic reads (which represent the majority of reads in the nanopore sequencing runs) might not be m⁶A-modified, but genomic reads might be modified, and we are not being able to identify these sites due to being a minority of the reads that are being analyzed. To test this, *NanoConsensus* analysis was performed on CHIKV-infected HEK293T versus CHIKV IVT samples, using reads that could be assigned unequivocally to either the genomic or subgenomic transcript as input to identify differentially modified regions (see "Methods"). Our results revealed no replicable sites of m⁶A modification in either genomic or subgenomic CHIKV RNAs (Fig. 4b and see also Supplementary Table 3).

Finally, as an additional control of our pipeline, we examined whether it could identify m⁶A modifications in a nuclear-replicating double-stranded DNA (dsDNA) virus. To this end, we analyzed publicly available DRS data from A549 cells (wild type and METTL3 knockout) infected with adenovirus serotype 5 (Ad5)[51]. Our results revealed 112 replicable sites that were identified across the Ad5 transcriptome (Supplementary Fig. 3a, b). Motif enrichment analysis was performed to validate these results, revealing that most of the identified sites contained a DRACH motif (Supplementary Fig. 3c), which is expected in METTL3-dependent m⁶A sites[15]. Altogether, our results indicate that CHIKV RNA (genomic and subgenomic transcripts) is either not m⁶A-modified, or modified at a very low stoichiometry, below the ~10% detection limit of the *NanoConsensus* algorithm[49].

## Depletion of the writer METTL3 or the eraser FTO do not affect CHIKV infection

If CHIKV RNA is m⁶A-modified, it would be expected that depletion of key components of the m⁶A machinery would affect CHIKV infection. To address this, we generated two stable shRNA knockdown HEK293T cell lines targeting either METTL3, the catalytic enzyme of the writing complex, or FTO, an eraser. We chose to deplete FTO over ALKBH5 because in HEK293T cells a significant proportion of FTO is found in the cytoplasm and thus this demethylase would have access to the viral RNA[20]. In contrast, no significant proportion of ALKBH5 is found in the cytoplasm[19]. Neither knockdown significantly affected expression of capsid protein, viral RNA levels, or viral titers (Fig. 5a, b). However, siRNA-mediated silencing of the cytoplasmic reader YTHDF1 resulted in increased CHIKV titers without affecting viral RNA or protein levels (Fig. 5c), suggesting that YTHDF1 suppresses a step of viral assembly or budding during CHIKV infection. This antiviral role might be mediated via direct interaction of YTHDF1 with the viral RNA. Alternatively, YTHDF1 depletion might interfere with cellular processes affecting CHIKV assembly. Other m⁶A readers, such as YTHDF2 or YTHDF3, were not assessed in this study, as it was recently shown that these proteins are functionally redundant[24] and in general the knockdown of these proteins results in a similar outcome of viral infection in multiple viruses[38]. Therefore, we only depleted YTHDF1 as a representative m⁶A reader. Next, we assessed whether CHIKV infection alters the location of endogenous METTL3, METTL14, WTAP, FTO, or YTHDF1 by immunofluorescence analyses. J2 antibody, which recognizes dsRNA structures[52], efficiently visualized the CHIKV RNA (Supplementary Fig. 4). If CHIKV RNAs were subjected to substantial METTL3-mediated m⁶A modification, one might expect the relocalization of this enzyme from the nucleus to the cytoplasm, due to the high abundance of CHIKV RNA[43]. In non-infected cells, as previously described[11,13,20,24], METTL3, METTL14, WTAP, and FTO had a predominantly nuclear localization while YTHDF1 was cytoplasmic in both HEK293T and U2OS cells (Supplementary Figs. 5 and 6, respectively). Importantly, none of these proteins changed location following CHIKV infection (Supplementary Figs. 5 and 6). Together, these results indicate that the main components of the m⁶A writing and erasing machinery do not affect CHIKV infection, and reciprocally, CHIKV infection does not alter the expression and localization of the m⁶A machinery.

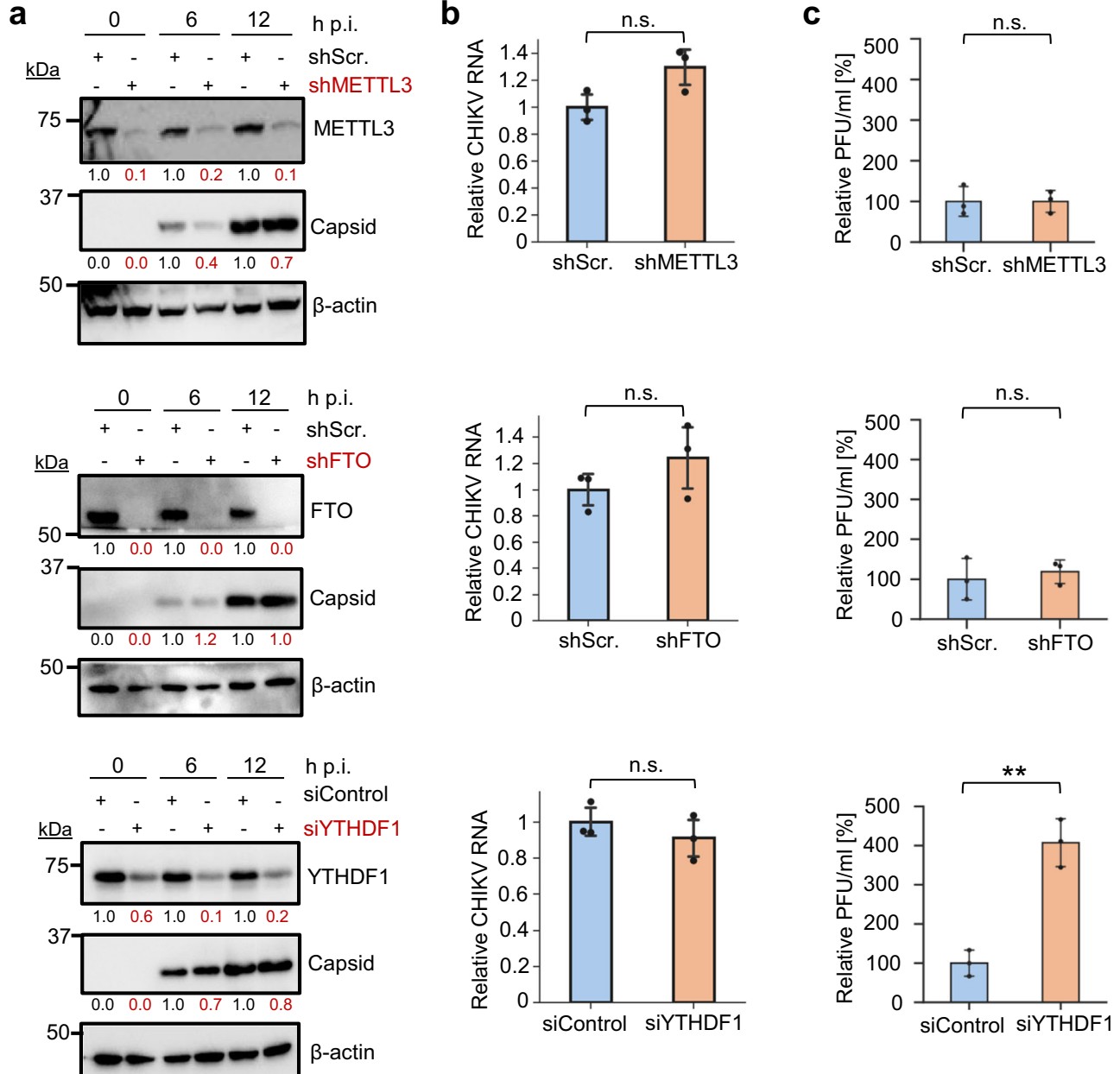

**Fig. 5 | Effect of depletion of METTL3, FTO, or YTHDF1 on CHIKV infection.**
Stable shRNA HEK293T cell lines were infected for 12 h at an MOI of 4. siRNA-treated (siControl and siYTHDF1) HEK293T cells were infected following 48 h of knockdown for 12 h at an MOI of 4. Scr. scramble, h p.i. hours post-infection. **a** Western blot quantification analyses are representative of 2 independent infections. β-actin is shown as a loading control. β-actin-normalized values from depleted samples, below each band, are shown relative to their controls. Uncropped blots can be found in Source Data. **b** Intracellular viral RNA levels were quantified by qRT-PCR and normalized against the housekeeping gene *GAPDH*. gRNA levels in depleted samples are shown relative to their corresponding controls. The bar chart shows mean values from 3 independent infections with the error bars showing SD. Source data are provided with this paper. **c** Supernatants collected at 12 h p.i. from CHIKV-infected control and knockdown cells were titered by plaque assay in HEK293T cells. The bar chart shows relative mean values from 3 independent replicates with the error bars showing SD. All statistical analyses were performed using a two-tailed *t*-test. n.s. not significant, **$p < 0.01$, *p* value = 0.0016. Source data are provided with this paper.

## m⁶A-Seq analysis of the DENV RNA genome shows a single m⁶A peak not confirmed by the SELECT technique

The absence of detectable m⁶A modification in CHIKV RNAs by antibody-independent techniques opened the possibility that the m⁶A modifications described in other exclusively cytoplasmic RNA viruses, all detected by antibody-dependent m⁶A-Seq, were also unreliable false positive sites that require further validation. Thus, we carried out similar experiments in DENV, reported to contain ten m⁶A peaks by m⁶A-Seq[39].

The DENV genome consists of a single 5′capped positive strand RNA (10.7 kb) from which all DENV proteins are translated (Fig. 6a). Following fragmentation of total DENV-infected HEK293T or Huh7 RNA into ~1 kb-long fragments, we could observe no significant m⁶A enrichment across any viral region by m⁶A-IP-qRT-PCR (Fig. 6b, c), despite the DENV RNA being in excess compared with the *SLC39A14* transcript (x150 times higher in HEK293T cells and x6 in Huh7 cells) (Fig. 6d). The percentage of recovery for all viral regions was <-0.4% in both cell lines, below the recovery of the non-modified *SLC39A14*

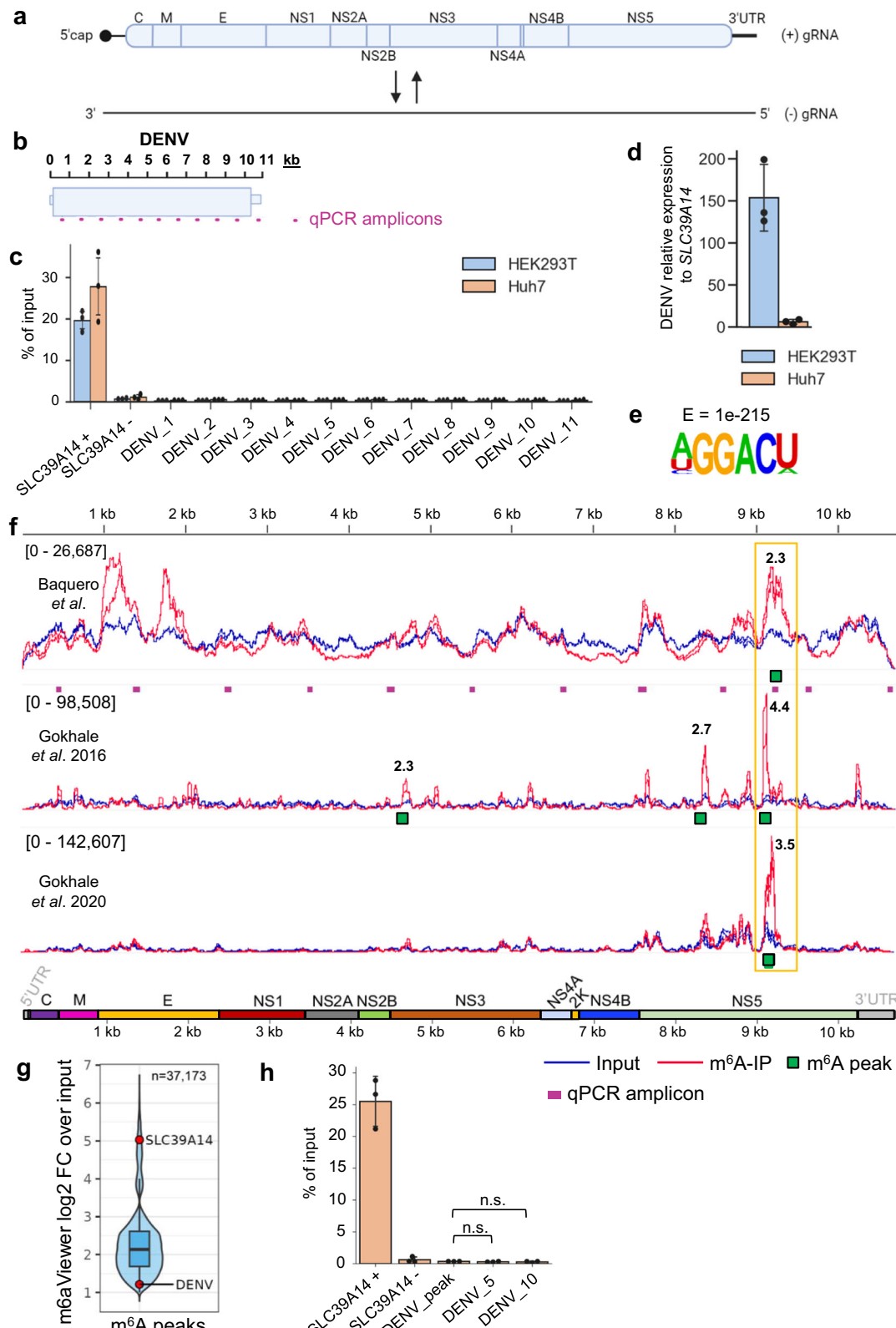

region (see Source Data Fig. 6c). In addition, we carried out m⁶A-Seq with ribodepleted RNA isolated from DENV-infected Huh7 cells. Peak-calling performed with m6aViewer detected 37,173 m⁶A cellular peaks conserved across both replicates (Supplementary Data 2). HOMER motif analysis of these peak regions revealed a DRACH motif to be the most significantly enriched (Fig. 6e). Next, we compared our m⁶A-Seq dataset with the two DENV (DENV2-NGC strain) publically available

m⁶A-Seq datasets performed on the same cell line[39,53]. Only one individual 300 nt-long peak was conserved between the three datasets in the NS5 region (Fig. 6f, yellow box), highlighting the high false positive rate and low reproducibility of antibody-based techniques for the detection of m⁶A modification[15,29,30,54]. Similar to the m⁶A peak we observed within our CHIKV datasets, this peak showed a mere 2.3-fold enrichment over input, significantly lower than the 4.4-fold median

**Fig. 6 | m⁶A-Seq analysis of DENV RNA reveals a single putative m⁶A peak.**
**a** Schematic diagram of the sense and antisense genomic RNA generated during DENV infection. **b** Schematic depiction of the tiled DENV qPCR amplicons used in m⁶A-IP-qRT-PCR analysis. **c** m⁶A-IP-qRT-PCRs performed with different DENV-infected cell lines using total RNA fragmented to ~1 kb. 11 primer sets (one for every kb: DENV_1 to DENV_11) were tiled along the DENV RNA. The bar chart shows mean values from 3 m⁶A-IPs across 3 independent infections with error bars showing SD. All cell lines were infected for 48 h at an MOI of 2. Source data are provided with this paper. **d** Intracellular viral RNA levels were quantified by qRT-PCR and normalized against the housekeeping gene *GAPDH*. For each cell line, viral RNA levels were expressed relative to *SLC39A14*. The bar chart shows mean values from 3 independent infections with error bars showing SD. All cell lines were infected for 48 h at an MOI of 2. Source data are provided with this paper. **e** Most significantly enriched motif identified in conserved m⁶A peaks across cellular RNA in DENV m⁶A-Seq data. HOMER software was used for motif analysis. **f** Genome browser tracks showing mapped DENV-2 (strain 16681) reads for input and m⁶A-IP samples. The reads are scaled to the same read depth against the viral genome using counts per million normalization. All biological replicates for each condition are displayed within each track. Conserved m6aViewer-called m⁶A peaks, which were also found by MACS2, are indicated. The fold change of m⁶A-IP/input, averaged from all replicates is shown above the called m⁶A peaks. Huh7 cells were infected for 48 h at an MOI of 3. Additional published DENV-2 m⁶A-Seq samples in the same cell line are shown as a comparison. Gokhale et al.[39] used DENV2-NGC strain at MOI of 2 (24 h p.i.) while Gokhale et al.[53] used DENV2-NGC strain at MOI of 1 (48 h p.i.). The 11 qPCR amplicons generated by the 11 primer sets (DENV_1 to DENV_11) are indicated. The qPCR amplicon spanning the conserved DENV m⁶A peak is also shown. **g** Violin and box plot of the log2 fold-change (log2FC) distribution of all 37,173 m6aViewer-called cellular peaks in DENV-infected Huh7 cells, conserved between two independent replicates. The *SLC39A14* and DENV peaks are indicated by red dots. The boxplot shows the median (middle line), 25–75th percentile values and 1.5x interquartile range, while the overlaid violin plot shows the full data distribution. **h** m⁶A-IP-qRT-PCRs performed to validate the putative m⁶A peak identified via m⁶A-Seq. Huh7 cells were infected for 48 h at an MOI of 2. Total RNA was fragmented to 100–200 nt. The bar chart shows mean values from 3 m⁶A-IPs from 3 independent infections with the error bars showing SD. DENV_5 and DENV_10 primers were used as negative control viral primers amplifying regions without any observed m⁶A enrichment according to our m⁶A-Seq data. n.s. not significant. ($p > 0.05$, using the two-tailed *t*-test). Source data are provided with this paper.

m⁶A peak enrichment of the cellular dataset and fell within the 5% most lowly enriched peaks (Fig. 6g). As a comparison, the known m⁶A site within the *SLC39A14* cellular transcript displayed a 33-fold enrichment, falling within the 5% most highly enriched peaks of the dataset (Fig. 6g), despite only containing <0.01% of mapped reads in the input, while viral reads constituted ~4.5% (Supplementary Table 1). We could not confirm this putative m⁶A peak via m⁶A-IP-qRT-PCR (using total RNA fragmented to ~100–200 nt) as we observed no significant m⁶A enrichment compared to non-modified regions (Fig. 6h). This suggested that it was either a false positive m⁶A peak or that the level of m⁶A modification at this region was extremely low.

To further assess the presence of putative m⁶A modifications, we used the highly sensitive SELECT method to interrogate the methylation status of the 13 DRACH motifs identified within the m⁶A peak (9114–9414 nt) in Huh7 and HEK293T cells. Only motif 8 (9291–9295 nt; AAACA) showed a significant reduction in the number of PCR cycles between the total RNA $\Delta C_T$ and the IVT $\Delta C_T$ (Fig. 7a and Supplementary Figs. 7 and 8). This would indicate the presence of an m⁶A at this motif. However, we observed that motif 8 resides within a recently identified conserved RNA loop, which is located within a similarly structured region in DENV-1, DENV-2, and DENV-4 serotypes[55] (Fig. 7b, c). We reasoned that this RNA loop structure might hinder Bst 2.0 DNA polymerase elongation in SELECT reactions resulting in the observed reduction in PCR cycles. Alternatively, this might also be caused by the presence of a different RNA modification. To address whether the reduction in PCR cycle number was specific to adenosine modification, we carried out a synonymous mutation using site-directed mutagenesis, which disrupted the DRACH motif (AAA<u>C</u>A > AAA<u>T</u>A) while preserving the structure of the RNA loop (Fig. 7d). Subsequently, we conducted SELECT using total RNA extracted from cells infected with the mutant virus (DENV-2 Mut) or DENV-2 Mut IVT RNA, both of which revealed a similar reduction in the number of PCR cycles (Fig. 7e), confirming that the DRACH in motif 8 is not m⁶A-modified.

Consistent with DENV RNA not being m⁶A-modified, depletion of METTL3 or FTO did not affect DENV infection. However, YTHDF1 depletion resulted in decreased DENV viral titers without affecting the viral RNA or protein levels (Supplementary Fig. 9). Moreover, DENV infection did not affect the localization of the host m⁶A machinery (Supplementary Fig. 10). As an example of quantification of the fluorescent signal, see Supplementary Fig. 11.

Taken together, these results demonstrate that CHIKV and DENV do not harbor m⁶A modifications that can be validated by antibody-independent techniques. In contrast, m⁶A modifications found in host RNAs and viral transcripts from DNA viruses that replicate in the nucleus are readily detected by all methods employed in this study, highlighting the importance of orthogonal antibody-independent validation of RNA modifications.

## Discussion

RNA modifications are central features in post-transcriptional regulation and are known to be dynamic upon viral infection[43,53,56]. Similarly, previous works have reported the presence of m⁶A modifications in distinct single-stranded cytoplasmic-replicating RNA viruses[38]. In this work, we have re-examined this latter point using a combination of orthogonal methods, including LC-MS/MS, m⁶A-Seq, SELECT, and nanopore DRS. Our results challenge the widely accepted idea that both cytoplasmic- and nuclear-replicating viruses undergo extensive m⁶A modification[38]. Our comprehensive analysis of the RNA genomes of two cytoplasmic-replicating viruses, CHIKV and DENV, found no evidence of m⁶A modifications in their genomes, despite previous reports indicating their presence[39,41]. It is worth noting that our data cannot rule out the putative presence of m⁶A modifications at such low stoichiometry that they cannot be detected by orthogonal approaches. m⁶A-Seq identified putative m⁶A modifications in both viruses, but these could not be validated by m⁶A-IP-qRT-PCR or orthogonal antibody-independent methods. The reason for the discrepancy between m⁶A-IP-qRT-PCR and m⁶A-Seq is not due to mis-mapped host reads to the CHIKV genome. The time of fragmentation is different between m⁶A-IP-qRT-PCR and m⁶A-Seq, but we believe this is not the reason for the discrepancy. It is more plausible that the bias is introduced in m⁶A-Seq samples. These go through a library preparation and RNA-seq analysis, which intrinsically is noisy and subjected to potential biases in amplification during library preparation or sequencing. The input control should account for most of these biases; however, the input library and the m⁶A-IP library are two distinct samples with vastly varied abundance of RNA fragments that may not behave similarly during RNA-seq. Because of this, to address putative m⁶A sites it is essential (1) to perform m⁶A-IP-qRT-PCR with internal controls within the IP sample and (2) carry out orthogonal approaches, particularly when m⁶A peaks show a low m⁶A enrichment. In agreement with m⁶A modification not playing a significant role in CHIKV and DENV viral cycles, CHIKV and DENV infections did not alter the location of the m⁶A machinery, and depletion of key components of the m⁶A writing and erasing machinery did not affect CHIKV or DENV infections.

The majority of viral m⁶A epitranscriptomic studies rely on m⁶A-Seq to locate m⁶A modifications[38]. However, this antibody-dependent technique has major drawbacks. Firstly, ~50% of detected m⁶A peaks have been reported to be false positives[30] and m⁶A-Seq has

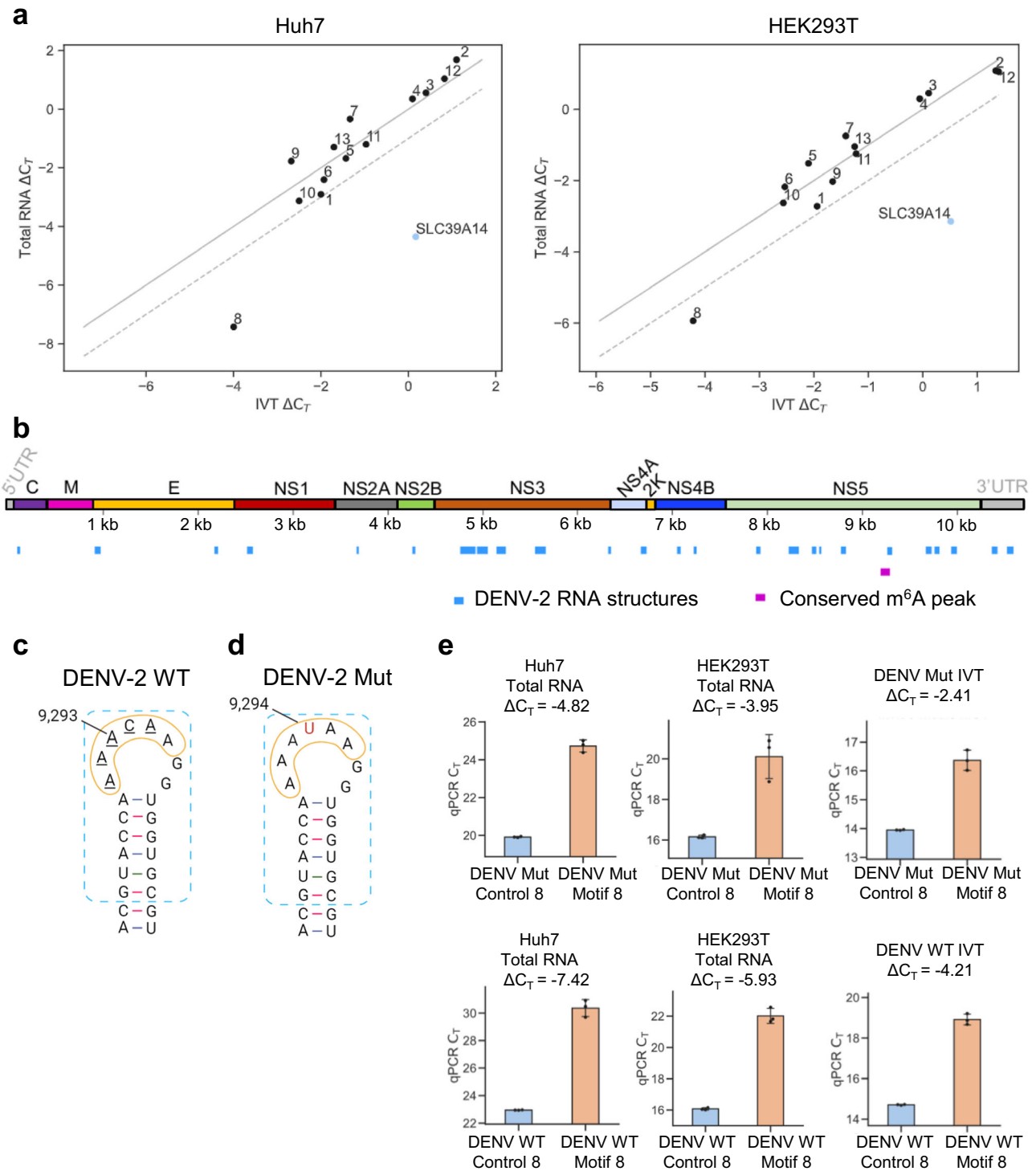

been shown to introduce nonspecific enrichment, causing recurrent false positives across hundreds of published m6A-Seq datasets[57]. Secondly, only ~30 to ~60% of m6A peaks are reproducible across different studies[29]. Thirdly, m6A-Seq does not provide accurate quantification of m6A levels at a specific site, and estimates are made by calculating the fold change of m6A-IP over input reads. Finally, m6A-Seq cannot discriminate between overlapping RNA isoforms. These drawbacks are diminished when highly m6A-modified transcripts are scrutinized, as in the case of nuclear-replicating viruses, such as Kaposi's sarcoma-associated herpesvirus, in which highly reproducible m6A peaks have been reported and validated by multiple independent groups[58–60]. Our m6A-Seq analyses identified m6A peaks in both the CHIKV and DENV

RNA genomes, suggesting putative m6A modifications sites. However, comprehensive validation using m6A-IP-qRT-PCR and the antibody-independent methods SELECT and nanopore DRS did not confirm the presence of any m6A modifications. Hepatitis C virus (HCV) has been reported to contain 19 m6A peaks via m6A-Seq[39]. Moreover, the authors generated a mutant virus in which the four identified DRACH motifs within an m6A peak in the E1 gene were mutated at once to study putative effects on the viral life cycle. Interestingly, the mutant virus displayed around a 3-fold increase in the viral titer when compared with the wild type virus without affecting HCV RNA replication. However, these interesting results lacked orthogonal validations with antibody-independent methods. Moreover, the observed phenotype

**Fig. 7 | SELECT and mutagenesis analyses of DENV RNA reveal no m⁶A modification. a** SELECT was performed using total RNA extracted from either Huh7 cells infected with DENV-2 (wild type) for 48 h at an MOI of 2 or from HEK293T cells infected for 96 h at an MOI of 0.5. Additionally, SELECT reactions were carried out with a ssDNA containing the modified DRACH motif in the *SLC39A14* RNA, as well as with DENV-2 in vitro transcribed (IVT) RNA. The threshold cycle difference of amplification ($\Delta C_T$) of total RNA versus IVT $\Delta C_T$ is shown for each motif (1–13). If total RNA $\Delta C_T$ = IVT $\Delta C_T$, values would align on the solid diagonal line. If there is a difference of −1 $C_T$ cycle between the total RNA $\Delta C_T$ and the IVT $\Delta C_T$, which would indicate the motif is m⁶A-modified, values would align on the diagonal discontinuous line. All experiments were performed using three technical replicates (separate SELECT reactions). All $C_T$ values from SELECT results can be found in Supplementary Figs. 7 and 8 and Source Data. **b** Previously elucidated conserved RNA elements in DENV-2 RNA[55] are shown in blue. The m⁶A peak (9114–9414 nt) conserved across all m⁶A-Seq studies is shown in purple. **c** Schematic diagram of the conserved RNA loop (yellow circle) in DENV-2 wild type (WT) identified by SHAPE-informed structure analysis[55], which contains DRACH motif 8 (AAACA, underlined nucleotides). The blue rectangle highlights the similar structure found in DENV-1, DENV-2, and DENV-4 serotypes[55]. The putative modified adenosine is at position 9293. This schematic diagram is adapted from Boerneke et al.[55]. **d** Schematic diagram of DENV-2 mutant (Mut) virus, carrying a mutation where the cytosine (at position 9294) within the DRACH motif 8 is changed to uridine (shown in red). **e** SELECT was performed using total RNA extracted from either Huh7 cells or from HEK293T cells (both infected with DENV-2 Mut for 96 h at an MOI of 0.5) to test the DENV-2 mutant motif 8. Additionally, SELECT reactions were carried out with DENV-2 mutant in vitro transcribed (IVT) RNA as control. All experiments were performed using three technical replicates (separate SELECT reactions). The bar chart shows mean values from three SELECT reactions with the error bars showing SD. $\Delta C_T$ values for SELECT reactions using total RNA extracted from DENV-2 wild type (WT)-infected cells and corresponding DENV-2 WT IVT are shown for comparison. Source data are provided with this paper.

may not have been directly linked to m⁶A-methylation but to alteration of the viral RNA structure and/or interacting host factors. In an additional manuscript[61] the authors describe by immunofluorescence that HCV infection increases the cytoplasmic signal of the m⁶A accessory protein WTAP in the cytosol but not that of METTL3. Our quantification of the WTAP immunofluorescent signal did not reveal any signal increase in the cytoplasm of CHIKV- or DENV-infected cells, however it is plausible that differences exist among different viruses. Altogether, it will be of great interest to confirm m⁶A modifications on HCV RNA by orthogonal methods.

Overestimation of m⁶A modifications when using low resolution antibody-dependent techniques is not unprecedented, as demonstrated for other RNA modifications. For example, studies on *N*4-acetylcytidine (ac⁴C) modifications showed that antibody-based mapping significantly over-detects this modification when compared to antibody-free methods[31]. Similarly, antibody-based studies on $N^1$-methyladenosine (m¹A) modifications estimated a m¹A/A ratio in mRNAs of ~0.02%[62], while single-nucleotide resolution antibody-dependent methods demonstrated that m¹A modifications in mRNAs are extremely rare[32,33]. Likewise, caution should be taken when estimating RNA modification levels in viral RNAs by mass spectrometry, as traces of host rRNA and tRNAs can significantly influence the results. For example, a study that relied solely on mass spectrometry analysis proposed that a myriad of different RNA modifications decorate viral RNA genomes, including m¹A in human immunodeficiency virus type 1 (HIV-1)[63]. However, subsequent antibody-independent m¹A deep-sequencing profiling at base-resolution showed that m¹A is absent from the genomic HIV-1 RNA[64].

If viral genomes or their transcripts are m⁶A-modified, one would expect such modifications to have an effect on viral infection. Consequently, several studies have combined m⁶A-Seq with depletion or overexpression of the m⁶A machinery to establish the presence of m⁶A within the RNA genomes of cytoplasmic-replicating viruses[38]. However, this might be misleading as these effects can be indirectly caused by changes in the host cell epitranscriptome. In our study, depletion of the writer METTL3 or the eraser FTO had no effect on CHIKV or DENV infection. However, depletion of the cytoplasmic reader YTHDF1 increased CHIKV and decreased DENV viral titers, without affecting the levels of intracellular viral RNA or protein. A previous study reported a similar YTHDF1-mediated effect on CHIKV viral titers and a direct interaction of YTHDF2 and YTHDF3 with CHIKV RNA[41]. Since YTH interactions with unmethylated RNAs have been described[2–4,58], it is plausible that YTHDF1 interacts with CHIKV and DENV RNA in an m⁶A-independent manner to affect virion assembly and/or exit. Alternatively, depletion of YTHDF1 might affect the expression of a host factor involved in these processes.

Our data indicate that m⁶A modification is not a general trait of viral RNA genomes. Interestingly, this finding aligns with early m⁶A studies in viruses, which utilized chromatography for m⁶A detection and demonstrated that only viruses with access to the nucleus exhibited internal m⁶A modification. For instance, the transcripts of polyoma simian virus 40 (SV40) and adenovirus, DNA viruses that replicate in the nucleus, were shown to be extensively modified[65,66]. These findings have been recently validated by different techniques and research groups[51,67]. Another notable study performed in the 1970s reported that the genome of influenza virus, an RNA virus that replicates in the nucleus, was also m⁶A-modified[68], a finding recently validated by PA-m⁶A-Seq[69]. In contrast, the transcripts of vaccinia virus, a dsDNA virus that replicates in the cytoplasm, did not harbor m⁶A modification[70].

Our findings highlight the critical importance of employing orthogonal validation methods and standardized controls when assessing the presence of m⁶A modification in the genomes of exclusively cytoplasmic viruses. To ensure the robustness of m⁶A detection, different strategies should be implemented, including antibody-independent techniques, along with the establishment of consistent positive and negative controls. Ideally, this should include described m⁶A modifications within host RNAs and viral IVT RNAs, respectively. By implementing such a comprehensive approach, we can mitigate the potential for conflicting results between studies[71–73] and, consequently, re-evaluate the presence of m⁶A modifications in other cytoplasmic-replicating RNA viruses.

## Methods

### Cell lines
HEK293T (ATCC; CRL-11268; female), Huh7 (a gift from Prof. Francis Chisari, The Scripps Research Institute) and U2OS (a gift from Prof. Wolfram Brune, Leibniz Institute of Virology) cells were cultured at 37 °C and 5% $CO_2$ in Dulbecco's modified Eagle's medium with glutamine (DMEM) (Gibco, 41966-052) supplemented with 10% (v/v) fetal bovine serum (FBS) (Sigma-Aldrich, F7524) and 1% (v/v) non-essential amino acids (Gibco, 11140-035). BHK-21 Clone 13 (ATCC number CCL-10) were cultured in Glasgow Minimum essential medium (GMEM) (Lonza, BE12-739F) supplemented with 10% (v/v) FBS (Sigma-Aldrich, F7524) and 10% (v/v) tryptose phosphate broth (BD Biosciences, 260300).

### Viruses and infection conditions
Stocks of CHIKV strain LR2006-OPY1 (GenBank: DQ443544, kindly provided by Prof. A. Merits, University of Tartu) and DENV-2 strain 16681 (GenBank NC_001474, kindly provided by Prof. R. Bartenschlager) were generated in BHK-21 cells and titered by standard plaque assay in HEK293T cells. All CHIKV infections were carried out at an MOI of 4 and for 12 h through the manuscript, unless otherwise stated. All DENV infections were carried out at an MOI of 2 for 48 h, unless otherwise stated. The viral inoculum was incubated for either 1 h

(CHIKV) or 2 h (DENV). For CHIKV and DENV infections carried out in HEK293T cells, Huh7, or U2OS cells, virus titers were determined by plaque assay using HEK293T cells, as both CHIKV and DENV viruses produced distinct and readily observable plaques in this cell type.

## Generation of in vitro transcribed (IVT) RNA and viral stocks

Plasmid DNA encoding either the CHIKV (pSP6-ICRES1) or DENV-2 strains (pFK-DVs) was linearized using FastDigest NotI (Thermo Fisher, FD0594) or XbaI (Thermo Fisher, FD0684), respectively. Linearized DNA was purified by phenol:chloroform:isoamyl alcohol (25:24:1 pH = 8.0) (Sigma-Aldrich, P2069) and used as template for in vitro transcription with mMessage mMachine SP6 kit (capped RNA; Thermo Fisher, AM1340). IVT reactions were incubated for 4 hr at 37 °C in a volume of 20 µl which included 2 µl of GTP. Following DNase treatment, IVT RNAs were purified with the RNeasy mini kit (Qiagen, 74106) and 0.5 µg were run on a denaturing formaldehyde gel to ensure integrity, while the remaining IVT RNA was stored at −80 °C.

Stocks of CHIKV were generated directly by infection of BHK-21 cells with a stock (gifted by A. Merits, University of Tartu) previously generated from electroporation of BHK-21 cells. CHIKV supernatant was harvested after 24 h of infection, at which point a clear cytopathic effect was observed. For production of DENV stocks, $6 \times 10^6$ BHK-21 cells were resuspended in 400 µl of cytomix (120 mM KCl; 0.15 mM $CaCl_2$; 10 mM $K_2HPO_4/KH_2PO_4$ pH 7.6; 25 mM Hepes pH 7.6; 5 mM EGTA pH 7.6; 5 mM $MgCl_2$) and electroporated with 7 µg of IVT in an electroporation cuvette (0.4 cm gap). Shortly before electroporation, freshly prepared ATP (2 mM) and glutathione (5 mM) were added to the cytomix solution. Cells were then pulsed at 270 V, 975 µF with a Gene Pulser Xcell system (Bio-Rad) and directly resuspended in growth medium (Glasgow Minimum essential medium (GMEM) (Lonza, BE12-739F) supplemented with 10% (v/v) FBS (Sigma-Aldrich, F7524) and 10% (v/v) tryptose phosphate broth (BD Biosciences). Sixteen hours post-electroporation, the media was changed. DENV-containing supernatant was collected after 72 h post-electroporation.

## Poly(A) + RNA-selection

Total RNA was extracted using TRIzol (Thermo Fisher, 15596018) according to the supplier's protocol. TURBO DNA-*free* Kit (Thermo Fisher, AM1907) was used to remove any contaminating DNA from RNA samples. Isolated RNA was further purified by ethanol precipitation and 30–40 µg of total RNA were subjected to poly(A)-selection with Dynabeads Oligo $(dT)_{25}$ (Thermo Fisher, 61002) according to the manufacturer's instructions.

## Quantitation of m⁶A modifications by LC-MS/MS analysis

Isolated poly(A) + RNA was firstly subjected to a clean-up and concentration step using the RNA clean and concentrator kit (Zymo Research, R1015). Then, 200 ng of poly(A) + RNA were digested with a nucleoside digestion mix (NEB, M0649S) at 37 °C for 1 h. Samples were further desalted using HyperSep Hypercarb SPE Spin Tips (Thermo Fisher, 60109-404) and run using an LTQ-Orbitrap XL mass spectrometer (Thermo Fisher Scientific, San Jose, CA, USA) coupled to an EASY-nLC 300 (Thermo Fisher [Proxeon], Odense, Denmark). Ribonucleosides were loaded directly onto the analytical column and were separated by reversed-phase chromatography using a 50-cm homemade column with an inner diameter of 75 µm, packed with 4 µm Hydro-RP 80 Å (Phenomenex, cat # 04A-4375), as previously described[74]. Chromatographic gradients started at 95% buffer A and 5% buffer B with a flow rate of 250 nl/min for 5 min and gradually increased to 20% buffer B and 80% buffer A in 40 min. After each analysis, the column was washed for 10 min with 20% buffer A and 80% buffer B. Buffer A consisted of 20 mM ammonium acetate pH 4.5. Buffer B was 95% acetonitrile and 5% 20 mM ammonium acetate pH 4.5.

The mass spectrometer was operated in positive ionization mode with nanospray voltage set at 2 kV and source temperature at 200 °C.

Full MS scans were set at 1 micro scans with a resolution of 60,000 and a mass range of $m/z$ 100–700 in the Orbitrap mass analyzer. A list of masses was defined for further fragmentation (Supplementary Table 5). Fragment ion spectra were produced via collision-induced dissociation (CID) at normalized collision energy of 35% and they were acquired in the ion trap mass analyzer. Isolation window was set to 2.0 $m/z$ and activation time of 10 ms. All data were acquired with Xcalibur software v2.1. Serial dilutions were prepared using commercial pure ribonucleosides (1–2000 pg/µl, Carbosynth, Toronto Research Chemicals) to establish the linear range of quantification and the limit of detection of each compound. See Supplementary Table 6 for the catalog numbers of commercial ribonucleosides. A mix of commercial ribonucleosides was injected before and after each batch of samples to assess the instrument stability and to be used as an external standard to calibrate the retention time of each ribonucleoside. Acquired data were analyzed with the Skyline software (v20.2.0.343) and extracted precursor areas of the ribonucleosides were used for quantification. The raw mass spectrometry data have been deposited to the MetaboLights repository[75] with the dataset identifier MTBLS6978.

## m⁶A-IP-qRT-PCR

For m⁶A-IPs, total RNA from infected HEK293T, Huh7, or U2OS cells was extracted using TRIzol (Thermo Fisher, 15596018) according to the supplier's protocol and putative DNA contaminants removed with TURBO DNA-*free* Kit (Thermo Fisher, AM1907). Isolated RNA was purified by ethanol precipitation and total RNA was fragmented with RNA fragmentation reagent (Thermo Fisher, AM8740) for either 3 min or 10 min at 70 °C to achieve ~1 kb-long fragments or 100–200 nt-long fragments, respectively. Fragmented RNA was ethanol-precipitated, re-suspended in 50 µl of RNase-free water and stored at −80 °C. For each m⁶A-immunoprecipitation (m⁶A-IP), 50 µl of slurry of Magna ChIP Protein G magnetic beads (Sigma-Aldrich, 16-662) were washed twice with IP/wash buffer 20 mM Tris HCl pH 7.4, 150 mM NaCl, and 0.1% NP-40 (v/v). Beads were resuspended in 100 µl of IP/wash buffer and coated with 5 µg of rabbit anti-m⁶A antibody (Active motif, 61495) for 45 min at room temperature with rotation. Beads were then washed 3x with IP/wash buffer and m⁶A-IPs were prepared by mixing the antibody-coated beads with 910 µl of IP/wash buffer, 35 µl of 0.5 M EDTA pH 8.0, 4 µl of murine RNase inhibitor (NEB, 174M0314S), and 40 µg (HEK293T and U2OS cells) or 20 µg (Huh7) of fragmented total RNA. In total, 1% input samples (10 µl from the total 1 ml m⁶A-IP mixture) were removed before immunoprecipitation and stored at −80 °C. m⁶A-IPs were incubated overnight at 4 °C with rotation. Beads were then washed 3x with IP/wash buffer. IP samples were further incubated with 126 µl of IP/wash buffer, 15 µl of 10% SDS (v/v) and 9 µl of PCR-grade proteinase K (20 mg/ml) (Thermo Fisher, EO0491) for 30 min at 55 °C. After incubation, 150 µl of the supernatant containing the RNA was transferred to a new microcentrifuge tube and 100 µl of IP/wash buffer was added to each sample. RNA was purified with TRIzol LS (Thermo Fisher, 10296010) and ethanol-precipitated together with 1.5 µl of RNA-grade glycogen (Thermo Fisher, R0551). Input samples were processed together with m⁶A-IPs from the proteinase K treatment onwards. Purified input and immunoprecipitated RNAs were resuspended in 11 µl of RNase-free water and reverse transcribed as described in the qRT-PCR section below. qPCR normalization was performed as previously described[58].

## Two-step quantitative reverse transcription PCR (qRT-PCR)

Total RNA from cells was extracted using TRIzol (Thermo Fisher, 15596018) according to the supplier's protocol and any contaminating DNA removed with TURBO DNA-*free* Kit (Thermo Fisher, AM1907). Reverse transcription reactions (20 µl) were carried out with Super-Script III reverse transcriptase (Thermo Fisher, 18080085) according to the manufacturer's instructions and containing 4 µl 5x first-strand

buffer, 1 μl of murine RNase inhibitor (NEB, 174M0314S), 1 μl of 10 mM dNTP mix (Thermo Fisher, 18427-013), 50 ng of random hexamers (Bioline, BIO-38028), and 1 μg of total RNA (in a total volume of 11 μl). For m⁶A-IP-qRT-PCR samples, the whole 11 μl of immunoprecipitated and corresponding input samples were reversed transcribed in this way. qPCR reactions (10 μl) consisted of 1x *Power* SYBR green PCR master mix (Thermo Fisher, 4368706), 0.5 μM of each primer, and 4.5 μl template cDNA. Cycling was performed in a QuantStudio 12 K Flex (Thermo Fisher). The cycling program included a hold stage of 50 °C for 2 min and 95 °C for 10 min, followed by 40 cycles of 95 °C for 15 s (denature step) and 60 °C for 1 min (anneal/extend step). After qPCR, a melting curve analysis was performed between 60 °C and 95 °C to confirm amplification of a single product. A list of all primers used in this study is provided as Supplementary Data 1. CHIKV_1 to CHIKV_11 primers were previously described[41].

## m⁶A-Seq

For CHIKV m⁶A-Seq experiments, poly(A) + RNA from CHIKV-infected HEK293T cells (12 h p.i. MOI of 4) was fragmented for 10 min at 70 °C with RNA fragmentation reagent (Thermo Fisher, AM8740). After fragmentation, the RNA was cleaned up through an ethanol precipitation step. At this stage, a sample of fragmented poly(A) + RNA was saved as input RNA for later use in cDNA library construction while 5 μg of fragmented poly(A) + RNA were used per m⁶A-IP which were carried out as described above. For DENV m⁶A-Seq experiments, total RNA was isolated, DNase-treated, ethanol-precipitated and fragmented with RNA fragmentation reagent for 10 min at 70 °C. Then, the RNA was ethanol-precipitated followed by a ribodepletion step using the human FFPE/degraded RNA riboPOOL kit (siTOOLs Biotech, dp-K012-000057) according to the supplier's protocol. Five μg of fragmented and ribodepleted RNA was used per each m⁶A-IP which was carried out as described above. RNA samples to be used in next-generation sequencing (NGS) were validated and quantified with a Bioanalyzer RNA 6000 Pico chip (Agilent Technologies, 5067-1511). Three to 5 ng of RNA from input and corresponding m⁶A-IPs were used for NGS library production following the protocol NEBNext Ultra II directional RNA library prep kit for Illumina (NEB, E7760L), treating samples as rRNA-depleted and fragmented RNAs. Libraries were validated individually with a high sensitivity D1000 TapeStation (Agilent Technologies, 5067-5585 and 5067-5584) and pooled in equimolar concentration for sequencing. The final pool was sequenced in a NextSeq 500 platform (Illumina) High Output generating 75 bp paired-end reads.

## Processing of raw deep-sequencing data and quality control

Raw reads were subjected to quality control using FastQC v0.11.9[76] prior to, and following, adapter trimming using cutadapt v4.1[77], which allowed removal of reads with low quality bases ($q < 20$), short lengths after trimming (<25 bp) and orphan read pairs.

## Next-generation sequencing data alignment

The resulting FASTQ files were mapped using STAR v2.7.6a[78] with parameters --outFilterType BySJout against a merger of the hg38 reference genome and the given viral genome, added as an additional contig.

## m⁶A-Seq data analysis

m⁶A peaks were called using m6aViewer v1.6.1[44] with default parameters and MACS2[46] v.2.2.7.1, using parameters -q 0.01 --nomodel --extsize 100 -B --SPMR --bdg --keep-dup all -f BAMPE. To call viral peaks with MACS2, the viral mapped reads were extracted and used as input, with the viral genome length used as the effective genome size parameter. R software was used to filter significant peaks, defined as showing >2-fold enrichment over input and a false discovery rate (FDR) < 5% across two biological replicates. Conserved viral peaks

found by both analyses were considered significant, with the m6aViewer-reported fold-change over input displayed within genome browser visualization plots. m6aViewer-called peaks were additionally extended 50 bases in each direction, from the site of enrichment, for easier visualization. Motif analysis across conserved m6aViewer-called m⁶A peaks was performed using HOMER[45] v4.11 with parameters -rna -len 5,6 -size 50, which identified motifs within 50 bases of sequence surrounding the peak center. Previously published m⁶A-Seq DENV-2 datasets[39,53] were analyzed against the DENV2_NGC FASTA & annotation (https://www.ncbi.nlm.nih.gov/nuccore/AF038403.1).

## SELECT m⁶A detection method

SELECT technique was carried out as previously described[34]. For the *SLC39A14* mRNA, we mined PA-m⁶A-Seq datasets[27] and selected a DRACH motif (genomic coordinate Chr8:22419678) that overlapped PA-m⁶A-IPs and our m⁶A-IPs (Supplementary Fig. 1a, box). Total RNA was extracted using TRIzol (Thermo Fisher, 15596018), DNase I-treated, and ethanol precipitated. Three μg of total RNA (HEK293T, CHIKV 12 h p.i. MOI 4) (HEK293T, DENV 96 h p.i. MOI 0.5), 400 ng of total RNA (Huh7, DENV 48 h p.i. MOI 2), and 50 ng of IVT RNA were used per SELECT reaction. For the *SLC39A14* mRNA, 50 ng of a 100 nt-long synthesized single stranded DNA (ssDNA) that contains the modified DRACH motif (purchased from IDT) was used (sequence details in Supplementary Data 1). The RNA or in vitro controls were mixed with 40 nM Up primer, 40 nM Down primer, and 5 μM dNTP in 17 μl 1x rCutSmart buffer (NEB, 174B6004S). The RNA and primers were then annealed by incubating as follows: 90 °C (1 min), 80 °C (1 min), 70 °C (1 min), 60 °C (1 min), 50 °C (1 min), and 40 °C (6 min). Then, 3 μl containing 0.01 U *Bst* 2.0 DNA polymerase (NEB, 174M0537S), 0.5 U SplintR ligase (NEB, 174M0375S), and 10 nmol ATP (NEB, 174P0756S) were added to the annealed products. The final reaction mixture (20 μl) was incubated at 40 °C for 20 min and denatured at 80 °C for 20 min. qPCR reactions (10 μl) consisted of 1x *Power* SYBR green PCR master mix (Thermo Fisher, 4368706), 100 nM of each SELECT primer and 1 μl neat of the final reaction mixture (except for the ssDNA *SLC39A14* which the final reaction mixture was diluted 1:1000). The cycling program included 95 °C for 10 min, followed by 40 cycles of 95 °C for 60 s (denature step) and 60 °C for 1 min (anneal/extend step). After qPCR, a melting curve analysis was performed between 60 °C and 95 °C to confirm amplification of single products. Primers used in SELECT qPCR measurement and up and down SELECT oligos are all listed in Supplementary Data 1. All SELECT oligos were purchased from IDT.

## Nanopore direct RNA sequencing library preparation

The following samples were sequenced in biological duplicates using direct RNA nanopore sequencing: (1) CHIKV in vitro transcribed (IVT) RNA; (2) poly(A)-selected RNA from HEK293T CHIKV-infected (12 h p.i., MOI of 4) cell lines (Supplementary Table 2). Direct RNA libraries were sequenced in independent flowcells or barcoded and pooled in a single flowcell using the barcoded DRS protocol, as previously described[79]. Briefly, each pair of A and B oligonucleotides was pre-annealed in annealing buffer (0.01 M Tris-Cl pH 7.5, 0.05 M NaCl) to a final concentration of 1.4 μM each in a total volume of 75 μl. The mixture was incubated at 94 °C for 5 min and slowly cooled down (−0.1 °C/s) to room temperature. RNA libraries for direct RNA Sequencing (SQK-RNA002) were prepared following the ONT Direct RNA Sequencing protocol version DRS_9080_v2_revI_14Aug2019 with half reaction for each library until the RNA Adapter (RMX) ligation step. RNAs were ligated to pre-annealed oligonucleotides using concentrated T4 DNA Ligase (NEB, M0202T), and the RNA was reverse transcribed using Maxima H Minus RT (Thermo Fisher, EP0752) following manufacturer's recommendations, without the heat inactivation step. The products were purified using 1.8x Agencourt RNAClean XP beads (Fisher Scientific, NC0068576) and washed

with 70% freshly prepared ethanol. In total, 50 ng of reverse transcribed RNA from each reaction was pooled and RMX adapter, composed of sequencing adapters with motor protein, was ligated onto the RNA:DNA hybrid and the mix was purified using 1x Agencourt RNAClean XP beads, washing with Wash Buffer (WSB) twice. The sample was then eluted in Elution Buffer (EB) and mixed with RNA Running Buffer (RRB) prior to loading onto a primed R9.4.1 flowcell, and ran on a MinION sequencing device. All information regarding the samples sequenced using nanopore sequencing, barcodes ligated to each sample, and sequencing throughput is described in Supplementary Table 2.

### Base-calling, demultiplexing and mapping of viral direct RNA sequencing runs

Raw fast5 reads from viral samples were analyzed using the MasterOfPores (MoP) version 2 Nextflow workflow[48] (Supplementary Table 2). Briefly, the *mop_preprocess* module was used to demultiplex the FAST5 reads using DeePlexiCon[79] with default parameters. Demuxed FAST5 were then base-called using Guppy 3.1.5 (Oxford Nanopore Technologies) with the model rna_r9.4.1_70bps_hac, and aligned to the human transcriptome from *Ensembl*, based on assembly GRCh38 (https://ftp.ensembl.org/pub/release-109/fasta/homo_sapiens/cdna/Homo_sapiens.GRCh38.cdna.all.fa.gz), supplemented with the CHIKV viral sequence using minimap2[80] v.2.17 with -uf -k14 -ax map-ont parameters (Supplementary Table 4).

### Analysis of genomic and subgenomic reads from viral direct RNA nanopore sequencing datasets

We then examined separately whether the reads coming from genomic or subgenomic RNAs would be m⁶A-modified. To this end, genomic and subgenomic reads from every viral DRS dataset (Supplementary Table 2) were extracted using an in-house Bash script (available at: https://github.com/novoalab/DRS_CHIKV_Analysis under the DOI:10.5281/zenodo.10555493[81]). We defined as "genomic" those reads whose length was equal or larger than 4517 nt were considered genomic transcripts, whereas reads whose alignment coordinates started between 7567–8067 nt and ended at position 11,700 nt or larger were subgenomic (Supplementary Table 2). We should note that the subgenomic RNA corresponds to the region 7567–11,313 of the CHIKV genome.

### Detection of RNA modifications in viral and human transcripts

Differential RNA modification detection was performed by pairwise comparison of viral samples to diverse control conditions (Supplementary Table 3) using the MasterOfPores (MoP2)[48] module *mop_mod* followed by *mop_consensus*. The first module predicts modified sites using 4 different softwares (EpiNano, Nanopolish, Nanocompore and Tombo). It was run using default parameters for *EpiNano* and *Nanopolish*, whereas *Nanocompore* was run with the addition of --downsample_high_coverage 5000. In the case of *Tombo*, samples with more than 100 K reads mapped to the viral genome were downsampled using tombo filter level_coverage with parameter --percent-to-filter 80 before running tombo level_sample_compare with default parameters and --multiprocess-region-size 50. Final consensus of differential RNA modifications was then generated using the module *mop_consensus*, which executes *NanoConsensus*. This algorithm uses as input the predictions of these 4 softwares run in *mop_mod* (*EpiNano, Nanopolish, Tombo* and *Nanocompore*). It was run with two sets of parameters: (1) default (--MZS_thr 5 --NC_thr 5) and (2) relaxed (--MZS_thr 3.75 --NC_thr 4). Only those sites identified in both replicates by *mop_consensus* were considered as true differentially modified sites (Supplementary Table 3). These configurations were used to identify differential modifications in CHIKV RNAs. For the analysis of adenovirus' transcriptome and human mRNAs only the default parameters were used (Supplementary Table 4).

### Publicly available deep-sequencing data

Data available from public repositories was downloaded and processed as above, except for use of the -f BAM MACS2 parameter for single-end m⁶A-Seq data. The following datasets were obtained from the NCBI Gene Expression Omnibus (GEO) as FASTQ files: m⁶A-Seq in DENV-2- infected Huh7 cells (24 h p.i.), two input and two m⁶A-IP replicates (GSM2203041, GSM2203042, GSM2203043, GSM2203044)[39]; m⁶A-Seq in DENV-2-infected Huh7 cells (48 h p.i.), three input and three m⁶A-IP replicates (GSM3755800, GSM3755801, GSM3755802, GSM3755803, GSM3755804, GSM3755805)[53]. Two PA-m⁶A-Seq peak-called datasets (GSM1326564, GSM1326565)[27] were downloaded as bed files and converted to hg38 coordinates using the UCSC liftOver tool. The following nanopore direct RNA sequencing runs were taken from the European Nucleotide Archive (ENA): A549 wild type and METTL3 KO Ad5-infected cells (ENA accession: PRJEB35652)[51].

Data resources used in this study are: hg38 genome FASTA (https://hgdownload.soe.ucsc.edu/goldenPath/hg38/bigZips/hg38.fa.gz), CHIKV genome FASTA & annotation (https://www.ncbi.nlm.nih.gov/nuccore/DQ443544) and DENV-2 genome (strain 16681) FASTA & annotation (https://www.ncbi.nlm.nih.gov/nuccore/158976983).

### Site-directed mutagenesis (SDM)

A single point mutation to mutate the DENV-2 motif 8 AAACA (9291–9295) into AAATA in the DENV-2 wild type plasmid (pFK-DVs) was generated using QuickChange XL site-directed mutagenesis kit (Agilent, 200517) according to the manufacturer's instructions. Primers for SDM are described in Supplementary Data 1. Two μl of the *Dpn I* digested amplification product was transformed into 25 μl of NEB turbo competent *E.coli* (NEB, C2984). The desired mutation was initially confirmed by Sanger sequencing (Genomics Core Facility (UPF, Spain), and later the integrity of both the whole wild type and mutated plasmids was confirmed via NGS-based plasmid DNA sequencing.

### NGS-based plasmid DNA sequencing

Plasmid DNA from either wild type dengue plasmid or mutated dengue plasmid were quantified using a High Sensitivity Range Qubit (Thermo Fisher). Library preparation was carried out with the Nextera XT DNA Library Prep kit (Illumina, FC-131-1096) following the manufacturer's instructions. A Bioanalyzer High Sensitivity (Agilent, 5067-4627) was used to verify the quality of the libraries and to prepare the pool, which was quantified using a specific qPCR for library adapters (Kappa Biosystems-Roche, KK4854). Sequencing was performed on a MiSeq System (Illumina) in 2 × 150 cycles in a flowcell nano.

### Plasmid DNA variant calling

Raw sequence reads underwent quality control and trimming using trimmomatic[82] (v0.36) to obtain high-quality reads. Trimmomatic parameters included a minimum quality score of 25 and a minimum length of 35. Adapter sequences were also removed during this process. The processed reads were aligned to the DENV-2 (strain 16681) reference genome using BWA aligner[83] (v0.7.17). Default parameters were used for alignment. The quality of the mapped reads was evaluated using FastQC[76] (v0.11.5). FastQC analysis encompassed various quality metrics such as read quality scores, GC content, and sequence duplication levels. Variant calling was performed using the Genome Analysis Toolkit (GATK)[84] (v4.1.8.1). GATK was employed to identify and analyze variants, including single nucleotide polymorphisms (SNPs) and insertions/deletions (INDELs). The variant calling analysis adhered to the recommended GATK best practices.

### shRNA knockdowns

Lentiviruses were generated by transfection of HEK293T cells seeded the day before in 6-well plates using a three-plasmid system. Per 6-well, 8 μl of lipofectamine 2000 (Thermo Fisher, 11668019) were used

together with 1.2 µg of pLKO.1 plasmid expressing the shRNA against the protein of interest, 0.65 µg of pVSV.G, and 0.65 µg psPAX2. pVSV.G, psPAX2, METTL3 shRNA (TRCN0000289812), and FTO shRNA (TRCN0000246247) were all a gift from Prof. A. Whitehouse (University of Leeds, UK)[58]. Scramble shRNA was a gift from Prof. David Sabatini (Addgene plasmid # 1864). 8 h post-transfection, medium was changed into 2 ml of DMEM (Gibco, 41966-052) supplemented with 10% (v/v) FCS. Two days post-transfection, 1.5 ml of viral supernatant was harvested, filtered through a 0.45 µm filter (Merck Millipore) and immediately used for transduction of $0.5 \times 10^6$ HEK293T cells that had been previously seeded the day before in a 6-well plate. Transductions were performed in the presence of 8 µg/ml polybrene (Sigma-Aldrich, TR-1003-G). Virus supernatant was removed at 8 h post-transduction and cells were maintained in fresh growth medium for 48 h before undergoing 3 µg/ml puromycin (Sigma-Aldrich, P4512-1MLX10) selection. Stable cell lines were generated after 4 days post-selection and were not subjected to more than 3–4 passages.

## siRNA knockdowns

In total, $0.15 \times 10^6$ HEK293T cells, seeded on 24-well plates the previous day, were transfected with either 100 nM of the specific YTHDF1 siRNA (SI00764715, Qiagen) or 100 nM siGENOME non-targeting siRNA (Dharmacon) using 2 µl of lipofectamine 2000 (Thermo Fisher, 11668019) per transfection. At 6–8 h post-transfection, the media was changed using normal growth media (DMEM) (Gibco, 41966-052) supplemented with 10% (v/v) fetal bovine serum (FBS) (Sigma-Aldrich, F7524) and 1% (v/v) non-essential amino acids (Gibco). Twenty-four hours post-transfection, cells were transfected again in the same way. All CHIKV infections were performed at 48 h post-siRNA transfection. When transfections were performed in a 12-well plate, these were performed in a similar manner as for 24-well plates, with the exception that the previous day $0.3 \times 10^6$ cells were seeded and 4 µl of lipofectamine 2000 were used.

## Western blotting

Cell pellets were lysed for 30 min in a lysis buffer (50 Mm Tris HCl pH 7.6, 150 mM NaCl, and 1% (v/v) NP-40) containing complete protease inhibitor cocktail (Sigma-Aldrich, 4693159001). Lysates were then centrifuged at maximum speed for 10 min and the supernatant was saved. Protein concentration was quantified with a BCA protein assay kit (Thermo Fisher, 23227) and lysates were then mixed 1:1 with 2x Laemmli buffer and boiled for 5 min at 95 °C. Lysates were then stored at −80 °C until needed. Protein samples were run on SDS-PAGE gels and transferred onto nitrocellulose membrane (GE healthcare, GE10600002) via wet transfer for 1.5 h at 100 V. Membranes were blocked with TBS + 0.1% (v/v) Tween 20 (TBST) and 5% (w/v) dried skimmed milk powder for 1 h, and then incubated overnight at 4 °C with relevant primary antibodies diluted in 5% (w/v) milk TBST. Membranes were washed 3x for 10 min with TBST and subsequently incubated for 1 h at room temperature with appropriate horseradish peroxidase-conjugated secondary antibodies (Sigma-Aldrich, NA934-1ML or NA931-1ML) diluted (1:10,000) in 5% (w/v) milk TBST. Membranes were washed again 3x for 10 min, treated with SuperSignal West Femto Maximum Sensitivity Substrate (Thermo Fisher, 34095) and chemiluminescence acquired with a ChemiDoc MP imaging system (Bio-Rad). Band quantification was performed with Image Lab Software (Bio-Rad).

Antibodies used in Western blotting were: anti-METTL3 (Abcam, ab195352, clone [EPR18810], 1:1000), anti-FTO (Abcam, ab126605, clone [EPR6894], 1:5000), anti-YTHDF1 (Proteintech, 17479-1-AP, 1:1000), anti-β-actin (Sigma-Aldrich, A5441-.2 ML, clone AC-15), anti-CHIKV capsid rabbit polyclonal antibody (1:5000), a gift from Prof. A. Merits (University of Tartu), anti-flavivirus group antigen (Novus Biologicals, NBP2-52709-0.2 mg, clone D1-4G2-4-15 (4G2), 1:500) and anti-DENV NS1 (GeneTex, GTX124280, 1:1000).

## Immunofluorescence

Cells were seeded the day before into 24-well plates containing sterile glass coverslips coated with poly-L-Lysine (Sigma-Aldrich, P1274-100MG). The seeding density per well was 150,000 cells for HEK293T and 50,000 cells for U2OS and Huh7 cells. Cells were fixed with 4% (v/v) formaldehyde (Sigma-Aldrich, F1635-500ML) diluted in PBS (Cultek, SH30028.02) for 10 min at room temperature. Immunofluorescence was then carried out as previously described[85] with the exception that in this study cells were incubated with DAPI for 10 min and mounted with Mowiol (Sigma-Aldrich, 324590). The following primary antibodies were used diluted in PBS with 1% (w/v) BSA (Sigma-Aldrich, A7906): anti-METTL3 (Abcam, ab195352, clone [EPR18810], 1:1000), anti-METTL14 (Sigma-Aldrich, HPA038002, 1:250), anti-WTAP (Proteintech, 60188-1-Ig, clone: 4A10G9, 1:250), anti-FTO (Abcam, ab126605, clone [EPR6894], 1:1000), anti-YTHDF1 (Proteintech, 17479-1-AP, 1:1000), anti-double-stranded RNA (clone J2) (Nordic-MUbio, 10010200, 1:2500), anti-CHIKV nsP1 (gift from Prof. A. Merits, 1:5000), anti-flavivirus group antigen (Novus Biologicals, NBP2-52709-0.2 mg, clone D1-4G2-4-15 (4G2), 1:1000) and anti-DENV NS1 (GeneTex, GTX124280, 1:1000). The following fluorescently conjugated secondary antibodies: Alexa Fluor 488 goat anti-rabbit IgG (Thermo Fisher, A-11008) and Alexa Fluor 568 goat anti-mouse IgG (Thermo Fisher, A-11004) were used at 1:500 dilution in PBS (Cultek, SH30028.02) with 1% (w/v) BSA (Sigma-Aldrich, A7906). Cells were imaged using a Leica SP5 inverted confocal laser scanning microscope and processed with Fiji (ImageJ)[86]. To quantify the WTAP signal from both the nucleus and the cytoplasm, we carried out image processing and analysis with Fiji software. DAPI staining was used to visualize the nucleus. The WTAP signal was enhanced throughout the entire image to delineate the cytoplasmic boundaries. CHIKV-infected cells were identified by nsP1 labeling and DENV-infected cells by NS1 labeling. Once the nucleus and cytoplasm boundaries were defined for each cell, the mean gray value was computed, representing the average gray value within the selected areas. This value was calculated by dividing the Intensity Density (IntDen) by the corresponding area. All raw values and quantified images can be accessed in Source Data.

## Reporting summary

Further information on research design is available in the Nature Portfolio Reporting Summary linked to this article.

## Data availability

The raw mass spectrometry data have been deposited to the MetaboLights repository with the dataset MTBLS6978. The m⁶A-Seq data and NGS-based plasmid DNA sequencing data from this study have been submitted to the NCBI GEO under accession number GSE231739. Base-called fast5 nanopore direct RNA sequencing data have been deposited at ENA under the accession PRJEB61652. NGS-based plasmid DNA sequencing data from this study have been deposited at ENA under the accession PRJEB66350. Source data are provided with this paper.

## Code availability

Detailed steps and in-house scripts used for the analysis of CHIKV direct RNA nanopore sequencing data are publicly available at GitHub (https://github.com/novoalab/DRS_CHIKV_Analysis) under the https://doi.org/10.5281/zenodo.10555493[81].

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

## Acknowledgements

This work was supported by funds from the Spanish Ministry of Science and Innovation (PID2022-136939OB-I00 funded by MICIN/AEI/10.13039/501100011033 and by "ERDF A way of making Europe" and PID2019106959RB-I00/AEI/10.13039/50110001103 to J.D.), Spanish Ministry of Economy, Industry and Competitiveness (MEIC) (PID2021-128193NB-100 to E.M.N.), the European Research Council (ERC-StG-2021 No. 101042103 to E.M.N.), an institutional "María de Maeztu" Programme for Units of Excellence in R&D (CEX2018-000792-M) and by the 2021 SGR 00176 grant from the Departament de Recerca i Universitats de la Generalitat de Catalunya. B.B.P. was the recipient of a Beatriu Pinós postdoctoral fellowship funded by the Secretary of Universities and Research (Generalitat de Catalunya) and by the Horizon 2020 programme of research and innovation of the European Union under the Marie Sklodowska-Curie grant agreement No. 801370. I.D.Y. and S.A.W. were supported by the Biotechnology and Biological Sciences Research Council, U.K. (BBSRC grant: BB/V00722X/1). A.D.T. and M.P.T. are supported by FPI Severo-Ochoa fellowships by the Spanish Ministry of Science and Innovation. We acknowledge the support of the MEIC to the EMBL partnership, Centro de Excelencia Severo Ochoa and CERCA Programme/Generalitat de Catalunya. We thank the Genomics Core Facility (UPF), in particular Núria Bonet Martín and Raquel Rasal Soteras for technical assistance with library preparation of m⁶A-Seq and plasmid DNA, and Marc Tormo for assisting with plasmid variant calling. The mass spectrometric analyses were performed in the CRG/UPF Proteomics Unit which is part of the Proteored, PRB3 and is supported by grant PT17/0019, of the PE I + D + i 2013–2016, funded by ISCIII and ERDF. We also thank Dr. Matthew Seddon for technical assistance for producing the data graphs. Figures 1a, c, d, 3a, 6a, b and 7c, d were created using Biorender for which we own a full licence.

## Author contributions

B.B.P., E.M.N., and J.D. conceived the project. B.B.P. designed and performed most molecular biology experiments and all viral infections. I.D.Y. analyzed m6A-Seq datasets. A.D.T. and O.B. performed the nanopore DRS processing and analyses. R.M. generated the libraries for nanopore DRS. M.P.T. produced DENV stocks and viral in vitro transcribed RNAs together with B.B.P. I.S. and S.A.W. supervised m6A-Seq data analyses. E.M.N. designed and supervised nanopore DRS analyses. B.B.P., I.D.Y., A.D.T., S.A.W., E.M.N. and J.D. evaluated the results. B.B.P. and J.D. wrote the manuscript with the contribution of all authors. All authors revised and commented on the manuscript.

## Competing interests

E.M.N. is a member of the Scientific Advisory Board of IMMAGINA Biotech. E.M.N. has received travel and accommodation expenses to speak at Oxford Nanopore Technologies conferences. A.D.T. and O.B. have received travel bursaries from ONT to present their work in conferences. Otherwise, the authors declare that the research was conducted in the absence of any commercial or financial relationships that could be construed as a conflict of interest. There are no more competing interests.
