## [Peer Review File · Nature Communications]

N⁶-methyladenosine modification is not a general trait of viral RNA genomesREVIEWER COMMENTS

Reviewer #1 (Remarks to the Author):

Previous studies have shown m6A modifications of the genomes of a number of RNA viruses that replicate in the cytoplasm, such as dengue virus, based on immunoprecipitation of RNA using antibodies against m6A subjected to next gen sequencing or RT-qPCR. Baquero-Pérez et al investigated m6A modification of the genomes of chikungunya and dengue viruses that replicate in human cells using various methods, including those based on anti-m6A antibodies or antibody independent. Their results suggest that the m6A peaks mapped to viral genomes identified by m6A-Seq analysis are not reproducible but are also undetectable by m6A-RT-qPCR. In all these experiments, the authors used a positive control (SLC39A14). Further, using Nanopore sequencing and SELECT, they found no evidence of m6A modifications in the virus genomes, while positive controls worked as expected. In addition, the authors showed that the m6A writer complex and the eraser are located mainly in the nucleus, while the tested reader was localized in the cytoplasm. These localisations did not change upon CHIKV or DENV infections. This is consistent with the lack of m6A modification of viral genomes in the cytoplasm, as access to m6A writers is obviously essential. Using the same approaches, the authors showed that the RNA transcripts of a DNA virus that replicates in the cell nucleus are m6A-modified. Besides, silencing the m6A machinery genes did not affect replication of CHIKV and DENV. I find that the results of this study are very important for RNA modification studies that go beyond virological studies and convincingly show that there is no m6A modification of the genome of RNA viruses that replicate in the cytoplasm. In general, the manuscript challenges the antibody-based detection of m6A modifications. The manuscript is written very well, and the conclusions are supported by the results. I only have minor typographical suggestions.

Line 33 and the rest of the manuscript: it is more common to use -Seq rather than -seq.

Line 99: ...we used

line 414 and 691: ...the medium was

line 488: is 4,5 meant to be 4.5?

Discussion: it is not customary to cite figures in the Discussion. I would remove them.

Reviewer #2 (Remarks to the Author):

Published work from various laboratories has reported the presence of N6 methylated adenosine (m6A) in the genomic and sub-genomic RNAs of RNA viruses believed to replicate exclusively in the cytoplasm of infected cells. In some instances, mutagenesis has been used to demonstrate the functional significance of those residues. That said, the frequency of these modifications in cytoplasmic RNA viruses and the enzymes responsible for installation remain controversial. Here the authors challenge the basic premise, at least in the context of Chikungunya virus (CHIKV) and dengue virus (DENV), representative arboviruses of biomedical importance. There is good evidence that both of these viruses transcribe their RNA in membrane-associated structures in the host cytoplasm using their own RNA-dependent RNA polymerases, which contrasts with host or nuclear DNA virus mRNAs and long non-coding RNAs that are methylated in the nucleus by well-studied factors associated with RNA polymerase II.

To address this, the authors perform a variety of studies on RNA isolated from three different human cell lines infected with CHIKV, a medically important alphavirus. The m6A-IP assay (Fig 1e) is particularly compelling and well-controlled. Interestingly, using an orthogonal approach (m6A-seq) they detect a single peak conforming to a cluster of classic methylation consensus sites for the primary host methyltransferase METTL3. However, the signal was relatively weak (compared to a host control) suggesting a relatively small number of RNA copies are actually methylated in

the samples tested. Both m6A-IP and m6A-seq are based on antibody capture, which has known issues with specificity. To explore this further, the authors use two antibody-independent methods, SELECT and direct RNA-sequencing by nanopore array. Again, these methods failed to detect m6A on CHIKV RNA and are well controlled. The possibility that m6A might be preferentially associated with the less abundant full-length genomic RNA rather than the hyperabundant sub-genomic RNAs is considered.

Despite the lack of evidence that CHIKV RNA is modified, the authors ask whether viral replication is impacted by the host m6A machinery and whether infection promotes detectable changes in the abundance or localization of key factors. The findings are uniformly negative. However, the reported ability of the cytoplasmic reader protein YTHDF1 to suppress CHIKV infectious virus levels is reproduced.

DENV is used as a different RNA virus (flavivirus) with multiple m6A sites reported previously (ref 39). A site within the NS5A gene is detected by m6A-seq and this correlates well with comparable data from others. However, the degree of enrichment was low and could not be recapitulated by m6A-IP-qPCR or by the SELECT amplification approach. To the authors, this low level of methylation is inconsistent with having a biological significance.

Overall, this is a rigorous and very interesting study that provokes a reconsideration of what constitutes adequate evidence for m6A modification of specific RNAs, whether they are from a virus or cellular RNA. The findings elegantly showcase continuing difficulties in quantifying the relative frequency at which individual RNA residues are modified and the importance of applying several methodologies to detect, and most importantly, verify putative-modified sites within a given RNA. As such the authors make a compelling argument for there being little to no m6A on the RNAs of the two cytoplasmic viruses they have tested. Whether this actually holds for all viruses that replicate in the cytoplasm is unknowable at present. The elephant(s) in the room are the published studies on hepatitis C virus, some of which are reanalyzed in Figure 6. Mutagenesis of the disputed sites produced a phenotype, and although not cited, there is published evidence implicating METTL3 and accessory factor WTAP in the modification of HCV RNAs. At a minimum, these seemingly contradictory findings should be tactfully acknowledged in the discussion.

MINOR SUGGESTIONS

Worthwhile stating why demethylase FTO was chosen rather than ALKBH5.

Fig 6f. The authors compare two datasets from Gokhale et al., analyzing DENV-infected cells at 24 and 48 hours after infection (hpi). At the earlier timepoint, there are five candidate peaks but only one is detected at both 24 and 48 hpi. The author's own data is from only 24 hours. To be rigorous, the possibility of changes in methylation over time needs to be addressed, perhaps using the SELECT assay.

Fig 7a. shows that only motif 8 is detected by SELECT-PCR to be potentially m6A modified. In the following experiments, however, the authors speculate that the presence of an RNA loop gave a false positive result. What about motif 1? It is very close to the discontinuous line in both cells, so it could be that it is modified at a low level. A time-course experiment could address whether there is a modification of this position earlier than 48 hpi that might be detectable by orthogonal means.

Page 12 DISCUSSION. Perhaps soften the statement "found no evidence of m6A modifications". The data doesn't rule out some modification at a very low frequency that is difficult to replicate across multiple orthogonal approaches.

Reviewer #3 (Remarks to the Author):

In this manuscript, the authors address a conundrum in the field of mRNA modifications. Several labs have reported m6A in the RNA genome of cytoplasmic viruses which never enter the nucleus. However, all known components of the m6A methyltransferase complex are localised to the nucleus. It is confusing how these viruses can become methylated when they are never in the same compartment in the methyltransferase complex. Here, using a large variety of orthogonal m6A-detection approaches, the authors show that DENV and CHIKV viruses lack m6A modifications in their genomes. The authors indicate that previous reports of m6A were due to false negatives, due to the techniques they used for measuring m6A.

Overall, the authors provide a convincing argument for the lack of m6A in CHIKV and DENV genomes. This is an important finding, firstly because it clarifies an ongoing conundrum in the field of whether these viruses have m6A in their genomes. Furthermore, this study highlights the importance of using orthogonal methods for mapping m6A. This study has important implications for the field, and will likely encourage others to re-evaluate m6A sites in other viruses more stringently.

The authors make use of m6A-seq, m6A-IP-qRT-PCR, SELECT, and direct RNA sequencing to measure m6A. Each of these individual methods have their shortcomings, and even data within this manuscript shows that each method is prone to false positives or biases. However, each of these methods rely on different principles, making them good orthogonal methods to validate m6A. By consolidating information from these methods, the authors convincingly demonstrate that previous reported m6A sites are false positive sites. However, there are a few inconsistencies which the authors should clearly address, which I have highlighted below.

Major concerns

1. In Fig 1, the authors replicate an experiment from Kim et al (Mol Cell 2020), where Kim et al observed CHIKV m6A sites. However, the authors do not see the same m6A sites which were previously observed. Can authors suggest why they achieved different results from Kim et al? Is this because of different methods of defining "enrichment", different clones of m6A antibodies, or other reasons? Can the data from this manuscript's experiment be considered more reliable than Kim et al's result?
2. The principle of m6A-IP-qRT-PCR and m6A-seq are very similar. I would expect that if the putative m6A peak in CHIKV is immunoprecipitated by m6A antibodies and sequenced by m6A-seq (Fig 2b), it would also be qRT-PCR'd (Fig 2d). Can the authors speculate on the reason for why the peak is only visible in m6A-seq and not via qRT-PCR? For example, is it mis-mapped to the CHIKV genome but is actually part of the host transcriptome? Is it due to different fragmentation conditions? Is there a reason to believe that m6A-IP-qRT-PCR or m6A-seq is more reliable? Understanding the reason for false negatives in m6A-seq may help identify guidelines for future analysis of m6A-seq.
3. The authors should be careful about interpreting the results of the shRNA experiments (Fig 5). Although it is tempting to conclude that YTHDF1 suppresses CHIKV viral assembly by interacting with CHIKV RNA (which the authors suggest in Line 232), it is also possible that shYTHDF1 interferes with other processes in the cell which later interfere with CHIKV assembly and does not interact with CHIKV RNA directly. I note that the authors highlight this in the discussion, but I encourage them to use the same caution when describing the result in the results section. This distinction of whether a knockdown of m6A writers/erasers/readers has a direct or indirect effect on the target of interest is important, as the misinterpretation that it is a direct effect often drives the idea that the target is methylated even when it is not.
4. In Fig 2c, the authors compare the fold-change of the m6A site over input in CHIKV (2.8x) to that of SLC39A14 (52x) to suggest that the CHIKV m6A site is "weak", as in it is lowly methylated or not methylated at all. However, this comparison is not appropriate as SLC39A14 might be a particularly highly methylated site. Instead, authors should show the range of the fold-change of all called m6A sites (or all m6A sites that are verified by a second method such as miCLIP), and

compare to the median instead.

Minor concerns

5. In Fig 2a, It is not clear in the caption if this is the consensus motif across all the mRNA, or just CHIKV RNA from infected cells.
6. In Figs 2d and 6g, the authors should indicate positions of the qPCR primers or the qPCR product relative to the m6A peaks.
7. In line 114, the sentence describing the ORFs of the gRNA is confusing. Are structural proteins expressed from both the sgRNA and gRNA second ORF, or only the sgRNA?
8. In Supp Fig 4, the authors should include a graph showing NanoConsensus scores of the Adenovirus transcript from the METTL3 KO cells.
9. In Supp Figs 5, 6, and 10, can the authors include quantification of nuclear:cytoplasmic signal in mock VS infected? It seems that there is more WTAP in the cytoplasm after infection with DENV (which has been previously reported with other types of viruses) but it is not clear from the images.

REVIEWER COMMENTS

Reviewer #1 (Remarks to the Author):

Previous studies have shown m6A modifications of the genomes of a number of RNA viruses that replicate in the cytoplasm, such as dengue virus, based on immunoprecipitation of RNA using antibodies against m6A subjected to next gen sequencing or RT-qPCR. Baquero-Pérez et al investigated m6A modification of the genomes of chikungunya and dengue viruses that replicate in human cells using various methods, including those based on anti-m6A antibodies or antibody independent. Their results suggest that the m6A peaks mapped to viral genomes identified by m6A-Seq analysis are not reproducible but are also undetectable by m6A-RT-qPCR. In all these experiments, the authors used a positive control (SLC39A14). Further, using Nanopore sequencing and SELECT, they found no evidence of m6A modifications in the virus genomes, while positive controls worked as expected. In addition, the authors showed that the m6A writer complex and the eraser are located mainly in the nucleus, while the tested reader was localized in the cytoplasm. These localisations did not change upon CHIKV or DENV infections. This is consistent with the lack of m6A modification of viral genomes in the cytoplasm, as access to m6A writers is obviously essential. Using the same approaches, the authors showed that the RNA transcripts of a DNA virus that replicates in the cell nucleus are m6A-modified. Besides, silencing the m6A machinery genes did not affect replication of CHIKV and DENV. I find that the results of this study are very important for RNA modification studies that go beyond virological studies and convincingly show that there is no m6A modification of the genome of RNA viruses that replicate in the cytoplasm. In general, the manuscript challenges the antibody-based detection of m6A modifications. The manuscript is written very well, and the conclusions are supported by the results. I only have minor typographical suggestions.

Line 33 and the rest of the manuscript: it is more common to use -Seq rather than -seq.

Line 99: ...we used

line 414 and 691: ...the medium was

line 488: is 4,5 meant to be 4.5?

Discussion: it is not customary to cite figures in the Discussion. I would remove them.

We have incorporated all the suggestions.

Reviewer #2 (Remarks to the Author):

Published work from various laboratories has reported the presence of N6 methylated adenosine (m6A) in the genomic and sub-genomic RNAs of RNA viruses believed to replicate exclusively in the cytoplasm of infected cells. In some instances, mutagenesis has been used to demonstrate the functional significance of those residues. That said, the frequency of these modifications in cytoplasmic RNA viruses and the enzymes responsible for installation remain controversial. Here the authors challenge the basic premise, at least in the context of Chikungunya virus (CHIKV) and dengue virus (DENV), representative arboviruses of biomedical importance. There is good evidence that both of these viruses transcribe their RNA in membrane-associated structures in the host cytoplasm using their own RNA-dependent RNA polymerases, which contrasts with host or nuclear DNA virus mRNAs and long non-coding RNAs that are methylated in the nucleus by well-studied factors associated with RNA polymerase II.

To address this, the authors perform a variety of studies on RNA isolated from three different human cell lines infected with CHIKV, a medically important alphavirus. The m6A-IP assay (Fig 1e) is particularly compelling and well-controlled. Interestingly, using an orthogonal approach (m6A-seq) they detect a single peak conforming to a cluster of classic methylation consensus sites for the primary host methyltransferase METTL3. However, the signal was relatively weak (compared to a host control) suggesting a relatively small number of RNA copies are actually methylated in the

samples tested. Both m6A-IP and m6A-seq are based on antibody capture, which has known issues with specificity. To explore this further, the authors use two antibody-independent methods, SELECT and direct RNA-sequencing by nanopore array. Again, these methods failed to detect m6A on CHIKV RNA and are well controlled. The possibility that m6A might be preferentially associated with the less abundant full-length genomic RNA rather than the hyperabundant sub-genomic RNAs is considered.

Despite the lack of evidence that CHIKV RNA is modified, the authors ask whether viral replication is impacted by the host m6A machinery and whether infection promotes detectable changes in the abundance or localization of key factors. The findings are uniformly negative. However, the reported ability of the cytoplasmic reader protein YTHDF1 to suppress CHIKV infectious virus levels is reproduced.

DENV is used as a different RNA virus (flavivirus) with multiple m6A sites reported previously (ref 39). A site within the NS5A gene is detected by m6A-seq and this correlates well with comparable data from others. However, the degree of enrichment was low and could not be recapitulated by m6A-IP-qPCR or by the SELECT amplification approach. To the authors, this low level of methylation is inconsistent with having a biological significance.

Overall, this is a rigorous and very interesting study that provokes a reconsideration of what constitutes adequate evidence for m6A modification of specific RNAs, whether they are from a virus or cellular RNA. The findings elegantly showcase continuing difficulties in quantifying the relative frequency at which individual RNA residues are modified and the importance of applying several methodologies to detect, and most importantly, verify putative-modified sites within a given RNA. As such the authors make a compelling argument for there being little to no m6A on the RNAs of the two cytoplasmic viruses they have tested. Whether this actually holds for all viruses that replicate in the cytoplasm is unknowable at present. The elephant(s) in the room are the published studies on hepatitis C virus, some of which are reanalyzed in Figure 6. Mutagenesis of the disputed sites produced a phenotype, and although not cited, there is published evidence implicating METTL3 and accessory factor WTAP in the modification of HCV RNAs. At a minimum, these seemingly contradictory findings should be tactfully acknowledged in the discussion.

The next paragraph has been added in the discussion section (page 13).

“Hepatitis C virus (HCV) has been reported to contain 19 m⁶A peaks via m⁶A-Seq (Gokhale *et al.* 2016). Moreover, the authors generated a mutant virus in which the four identified DRACH motifs within a peak in the E1 gene were mutated at once to study putative effects on the viral life cycle. Interestingly, the mutant virus displayed around a 3-fold increase in the viral titer when compared with the wild type virus without affecting HCV RNA replication. However, these interesting results lacked orthogonal validations with antibody-independent methods. Moreover, the observed phenotype may not have been directly linked to m⁶A-methylation but to alteration of the viral RNA structure and/or interacting host factors. In an additional manuscript (Sacco *et al.* 2022, *Journal of Virology*) the authors describe by immunofluorescence that HCV infection increases the cytoplasmic signal of the m⁶A accessory protein WTAP in the cytosol but not that of METTL3. Our quantification of the WTAP immunofluorescent signal did not reveal any signal increase in the cytoplasm of CHIKV- or DENV-infected cells, however it is plausible that differences exist among different viruses. Altogether, it will be of great interest to confirm m⁶A modifications on HCV RNA by orthogonal methods.”

The quantitative analyses of WTAP immunofluorescent signals have been included as Fig. S11 and the raw data added to Source Data (Fig. S11).

MINOR SUGGESTIONS

Worthwhile stating why demethylase FTO was chosen rather than ALKBH5.

We chose to deplete FTO over ALKBH5 because in HEK293T cells (where we performed shRNA-mediated depletions) a significant proportion of FTO is found in the cytoplasm and thus this demethylase would have access to the viral RNA (Wei, J. *et al. Differential m6A, m6Am, and m1A Demethylation Mediated by FTO in the Cell Nucleus and Cytoplasm. Molecular Cell* 71, 973-985.e975 (2018)). In contrast, no significant proportion of ALKBH5 is found in the cytoplasm (Thalhammer, A. *et al. Human AlkB Homologue 5 Is a Nuclear 2-Oxoglutarate Dependent Oxygenase and a Direct Target of Hypoxia-Inducible Factor 1 α (HIF-1 α). PLOS ONE* 6, e16210 (2011)).

We have now included this justification in the manuscript (results section, page 9).

Fig 6f. The authors compare two datasets from Gokhale *et al.*, analyzing DENV-infected cells at 24 and 48 hours after infection (hpi). At the earlier timepoint, there are five candidate peaks but only one is detected at both 24 and 48 hpi. The author's own data is from only 24 hours. To be rigorous, the possibility of changes in methylation over time needs to be addressed, perhaps using the SELECT assay.

While revisiting the five conserved m6A-viewer called peaks at 24h p.i. in the 2016 Gokhale *et al.* dataset, we realized that we had filtered conserved m6A-viewer called peaks to retain those which overlap with MACS2-called peaks from any replicate, instead of those only found in both MACS2-called replicates. We have corrected this error and updated Fig. 6f accordingly, which now reflects that only 3 of the 5 m6A-viewer-called peaks were conserved between both analyses and both replicates.

The life cycle of DENV is around 24 hours. Our m⁶A-Seq data is not from 24 hr p.i. but from 48 hr p.i. in Huh7 cells. This timing was purposely chosen since the DENV viral reads at 24 hr p.i. (\approx 1% viral reads) in Huh7 cells is approximately 10-fold lower than at 48 hr p.i. (\approx 10% viral reads) (See supplementary table 3, m6A-seq mapping statistics). These extremely low DENV RNA levels significantly increase the chances of false positive peaks.

A new bioinformatic analysis that we have now carried out shows that the 2.3-fold enrichment of m⁶A-IP/input in the conserved DENV peak at 48 hr p.i. is significantly lower than the 4.4-fold median m⁶A peak enrichment of the cellular Huh7 dataset (new Fig. 6g). Notably this viral peak was among the lowest 5% enriched peaks, strongly suggesting that it is a false positive as shown by m⁶A-IP-qPCR and SELECT. Similarly, the CHIKV peak, which was within the 5% most lowly enriched peaks with a 2.8-fold enrichment (new Fig. 2d), was a false positive peak.

Considering (i) that the levels of DENV RNA are extremely low at 24 hr p.i., (ii) that the two viral peaks detected at 24 hr p.i. in the dataset of Gokhale *et al.* have a 2.3- and 2.7-fold enrichment, an enrichment shown to be linked to false positives, and (iii) that these peaks were not detected at 48 hr p.i., a time at which two infection cycles have been completed, the two peaks detected by Gokhale *et al.* at 24 hr p.i. are most likely false positives.

Fig. 2d (Left panel) Violin and box plot of the log₂ fold-change (log₂FC) distribution of all 23,539 m⁶AViewer-called cellular peaks in CHIKV-infected HEK293T cells, conserved between two independent replicates. The *SLC39A14* and CHIKV peaks are indicated by red dots. The boxplot shows the median (middle line), 25-75th percentile values and 1.5x interquartile range, while the overlaid violin plot shows the full data distribution.

Fig. 6g (Right panel) Violin and box plot of the log₂ fold-change (log₂FC) distribution of all 37,173 m⁶AViewer-called cellular peaks in DENV-infected Huh7 cells, conserved between two independent replicates. The *SLC39A14* and DENV peaks are indicated by red dots. The boxplot shows the median (middle line), 25-75th percentile values and 1.5x interquartile range, while the overlaid violin plot shows the full data distribution.

Fig 7a. shows that only motif 8 is detected by SELECT-PCR to be potentially m⁶A modified. In the following experiments, however, the authors speculate that the presence of an RNA loop gave a false positive result. What about motif 1? It is very close to the discontinuous line in both cells, so it could be that it is modified at a low level. A time-course experiment could address whether there is a modification of this position earlier than 48 hpi that might be detectable by orthogonal means.

We have now carried out a SELECT experiment in Huh7 cells infected for 24 hr with DENV (MOI of 2). The motif 1 is just on the verge of the cut-off, but clearly differs from the signal seen for motif 8 or the *SLC39A14* RNA motif.

Raw C_T values are shown below:

SELECT DENV (wild type) Huh7 cells 24h p.i.

CT value	Oligo	RNA Type	Motif interrogated	Mean CT value	ΔCT between control and m6A motif oligos
18.37356	slc control	Total RNA	slc	17.55221176	
17.51441	slc control	Total RNA	slc		
16.76867	slc control	Total RNA	slc		
20.50066	slc m6A motif	Total RNA	slc	20.88507907	-3.332867304
21.06166	slc m6A motif	Total RNA	slc		
21.09292	slc m6A motif	Total RNA	slc		
13.58815	slc control	slc ssDNA	slc	13.46720219	
13.76562	slc control	slc ssDNA	slc		
13.04784	slc control	slc ssDNA	slc		
12.17155	slc m6A motif	slc ssDNA	slc	12.359677	1.10752519
12.4853	slc m6A motif	slc ssDNA	slc		

12.42218	slc m6A motif	slc ssDNA	slc		
21.23156	Control 1	Total RNA	motif1	21.24807167	
21.20832	Control 1	Total RNA	motif1		
21.30433	Control 1	Total RNA	motif1		
23.84824	m6A motif 1	Total RNA	motif1	23.88813082	-2.640059153
23.88164	m6A motif 1	Total RNA	motif1		
23.93452	m6A motif 1	Total RNA	motif1		
11.05762	Control 1	DENV WT IVT	motif1	10.75856813	
10.64636	Control 1	DENV WT IVT	motif1		
10.57172	Control 1	DENV WT IVT	motif1		
12.36581	m6A motif 1	DENV WT IVT	motif1	12.36738141	-1.608813286
12.31587	m6A motif 1	DENV WT IVT	motif1		
12.42047	m6A motif 1	DENV WT IVT	motif1		
23.94765	Control 8	Total RNA	motif8	23.9213473	
24.03713	Control 8	Total RNA	motif8		
23.77927	Control 8	Total RNA	motif8		
31.33869	m6A motif 8	Total RNA	motif8	30.08853531	-6.167188009
29.21252	m6A motif 8	Total RNA	motif8		
29.71439	m6A motif 8	Total RNA	motif8		
14.24721	Control 8	DENV WT IVT	motif8	14.2038517	
14.2596	Control 8	DENV WT IVT	motif8		
14.10475	Control 8	DENV WT IVT	motif8		
18.26181	m6A motif 8	DENV WT IVT	motif8	18.58400536	-4.380153656
18.8503	m6A motif 8	DENV WT IVT	motif8		

18.6399	m6A motif 8	DENV WT IVT	motif8		
---------	-------------	-------------	--------	--	--

Page 12 DISCUSSION. Perhaps soften the statement “found no evidence of m6A modifications”. The data doesn’t rule out some modification at a very low frequency that is difficult to replicate across multiple orthogonal approaches.

We have modified the text accordingly, adding an extra sentence (discussion, page 12):

Our comprehensive analysis of the RNA genomes of two cytoplasmic-replicating viruses, CHIKV and DENV, found no evidence of m⁶A modifications in their genomes, despite previous reports indicating their presence^{39, 41}. It is worth noting that our data cannot rule out the putative presence of m6A modifications at such low stoichiometry that they cannot be detected by orthogonal approaches.

Reviewer #3 (Remarks to the Author):

In this manuscript, the authors address a conundrum in the field of mRNA modifications. Several labs have reported m6A in the RNA genome of cytoplasmic viruses which never enter the nucleus. However, all known components of the m6A methyltransferase complex are localised to the nucleus. It is confusing how these viruses can become methylated when they are never in the same compartment in the methyltransferase complex. Here, using a large variety of orthogonal m6A-detection approaches, the authors show that DENV and CHIKV viruses lack m6A modifications in their genomes. The authors indicate that previous reports of m6A were due to false negatives, due to the techniques they used for measuring m6A.

Overall, the authors provide a convincing argument for the lack of m6A in CHIKV and DENV genomes. This is an important finding, firstly because it clarifies an ongoing conundrum in the field of whether these viruses have m6A in their genomes. Furthermore, this study highlights the importance of using orthogonal methods for mapping m6A. This study has important implications for the field, and will likely encourage others to re-evaluate m6A sites in other viruses more stringently.

The authors make use of m6A-seq, m6A-IP-qRT-PCR, SELECT, and direct RNA sequencing to measure m6A. Each of these individual methods have their shortcomings, and even data within this manuscript shows that each method is prone to false positives or biases. However, each of these methods rely on different principles, making them good orthogonal methods to validate m6A. By consolidating information from these methods, the authors convincingly demonstrate that previous reported m6A sites are false positive sites. However, there are a few inconsistencies which the authors should clearly address, which I have highlighted below.

Major concerns

1. In Fig 1, the authors replicate an experiment from Kim et al (Mol Cell 2020), where Kim et al observed CHIKV m6A sites. However, the authors do not see the same m6A sites which were previously observed. Can authors suggest why they achieved different results from Kim et al? Is this because of different methods of defining “enrichment”, different clones of m6A antibodies, or other

reasons? Can the data from this manuscript's experiment be considered more reliable than Kim et al's result?

We note several differences between the conditions employed in our study and those used by Kim *et al.* Firstly, Kim's study carried out RNaseT1 treatment to fragment the RNA, a practice not typically utilized in m⁶A-IP experiments, indeed we are not aware of any other m⁶A-IP published work where this enzyme was used for RNA fragmentation. Secondly, Kim's study used a different antibody (Rabbit polyclonal from Synaptic systems) for immunoprecipitation to ours (Rabbit polyclonal from Active motif), and finally, Kim's study displayed the m⁶A enrichment relative to an IgG control, in which two antibody specificities are compared for different viral RNA regions. Instead, we displayed the % of input recovered which is more robust to compare m⁶A enrichment directly between different studies.

The lack of overlapping m⁶A-modified regions in viral RNAs between different studies is not unique to this study, indeed it is quite common. For example, this the case in the two studies performed with Zika virus (Gokhale *et al.* 2016 and Lichinchi *et al.* 2016) or the three studies performed with HIV-1 (Lichinchi *et al.* 2016, Tirumuru *et al.* 2016 and Kennedy *et al.* 2016). All these studies relied on m⁶A-seq or PA-m⁶A-seq detection. This is why orthogonal approaches are essential tools to confirm putative m⁶A sites and to reduce false positive signals, as we propose in our study.

2. The principle of m⁶A-IP-qRT-PCR and m⁶A-seq are very similar. I would expect that if the putative m⁶A peak in CHIKV is immunoprecipitated by m⁶A antibodies and sequenced by m⁶A-seq (Fig 2b), it would also be qRT-PCRed (Fig 2d). Can the authors speculate on the reason for why the peak is only visible in m⁶A-seq and not via qRT-PCR? For example, is it mis-mapped to the CHIKV genome but is actually part of the host transcriptome? Is it due to different fragmentation conditions? Is there a reason to believe that m⁶A-IP-qRT-PCR or m⁶A-seq is more reliable? Understanding the reason for false negatives in m⁶A-seq may help identify guidelines for future analysis of m⁶A-seq.

The reason for the discrepancy between m⁶A-IP-qRT-PCR and m⁶A-Seq is not due to mis-mapped host reads to the CHIKV genome. The time of fragmentation is different between m⁶A-IP-qRT-PCR and m⁶A-Seq, but we believe this is not the reason for the discrepancy. It is more plausible that the bias is introduced in m⁶A-Seq samples. These go through a library preparation and RNA-seq analysis, which intrinsically is noisy and subjected to potential biases in amplification during library preparation or sequencing. The input control should account for most of these biases; however, the input library and the m⁶A-IP library are two distinct samples with vastly varied abundance of RNA fragments that may not behave similarly during RNA-seq. Because of this, to address putative m⁶A sites it is essential (i) to perform m⁶A-IP-qRT-PCR with internal controls within the IP sample and (ii) carry out orthogonal approaches, particularly when m⁶A peaks show a low m⁶A enrichment.

This important point is now commented on in the discussion section (page 12).

3. The authors should be careful about interpreting the results of the shRNA experiments (Fig 5). Although it is tempting to conclude that YTHDF1 suppresses CHIKV viral assembly by interacting with CHIKV RNA (which the authors suggest in Line 232), it is also possible that shYTHDF1 interferes with other processes in the cell which later interfere with CHIKV assembly and does not interact with CHIKV RNA directly. I note that the authors highlight this in the discussion, but I encourage them to use the same caution when describing the result in the results section. This distinction of whether a knockdown of m⁶A writers/erasers/readers has a direct or indirect effect on the target of interest is important, as the misinterpretation that it is a direct effect often drives the idea that the target is methylated even when it is not.

We thank the reviewer for pointing this out and we have modified the text in the results section (page 9) accordingly.

This antiviral role might be mediated via direct interaction of YTHDF1 with the viral RNA. Alternatively, YTHDF1 depletion might interfere with cellular processes affecting CHIKV assembly.

4. In Fig 2c, the authors compare the fold-change of the m6A site over input in CHIKV (2.8x) to that of SLC39A14 (52x) to suggest that the CHIKV m6A site is “weak”, as in it is lowly methylated or not methylated at all. However, this comparison is not appropriate as SLC39A14 might be a particularly highly methylated site. Instead, authors should show the range of the fold-change of all called m6A sites (or all m6A sites that are verified by a second method such as miCLIP), and compare to the median instead.

The requested analyses are now included in the results section (page 6 and 10 and Fig. 2d and Fig. 6g). These figures compare the range of the fold-change of all called m⁶A cellular sites in our HEK293T (CHIKV) and Huh7 (DENV) datasets. The analyses revealed that (i) the *SLC39A14* RNA is highly methylated, and therefore a suitable and robust positive control RNA and (ii) both viral peaks (CHIKV and DENV) lie significantly lower than the median fold enrichment of cellular peaks, strongly suggesting that they are false positive peaks.

The following highlighted text has been added on page 6:

“In CHIKV RNA, a single peak with 2.8-fold enrichment over input (**Fig. 2b**) was the only significant peak detected by two widely used m⁶A peak-calling programs, m6aViewer and MACS2⁴⁶ (**Supplementary table 2**). In contrast, the known m⁶A site within the *SLC39A14* cellular transcript fell within the top 5% most highly enriched peaks with a 52-fold enrichment (**Fig. 2c and 2d**), despite having much lower abundance than CHIKV RNA (50% of all input reads were viral while < 0.01% mapped to *SLC39A14*, **Supplementary Table 3**). The 2.8-fold enrichment seen at the viral peak was surprisingly low when compared with the 10.3-fold median m⁶A peak enrichment of the cellular dataset and fell within the 5% most lowly enriched peaks (**Fig. 2d**).”

The following text has been added on page 10:

“Similar to the m⁶A peak we observed within our CHIKV datasets, this peak showed a mere 2.3-fold enrichment over input, significantly lower than the 4.4-fold median m⁶A peak enrichment of the cellular dataset and fell within the 5% most lowly enriched peaks (**Fig. 6g**). As a comparison, the known m⁶A site within the *SLC39A14* cellular transcript displayed a 33-fold enrichment, falling within the 5% most highly enriched peaks of the dataset (**Fig. 6g**).”

Fig. 2d (Left panel) Violin and box plot of the log₂ fold-change (log₂FC) distribution of all 23,539 m⁶AViewer-called cellular peaks in CHIKV-infected HEK293T cells, conserved between two independent replicates. The *SLC39A14* and CHIKV peaks are indicated by red dots. The boxplot shows the median (middle line), 25-75th percentile values and 1.5x interquartile range, while the overlaid violin plot shows the full data distribution.

Fig. 6g (Right panel) Violin and box plot of the log₂ fold-change (log₂FC) distribution of all 37,173 m⁶AViewer-called cellular peaks in DENV-infected Huh7 cells, conserved between two independent replicates. The *SLC39A14* and DENV peaks are indicated by red dots. The boxplot shows the median (middle line), 25-75th percentile values and 1.5x interquartile range, while the overlaid violin plot shows the full data distribution.

Minor concerns

5. In Fig 2a, It is not clear in the caption if this is the consensus motif across all the mRNA, or just CHIKV RNA from infected cells.

We refer to the consensus motif across all the mRNA. The figure legend is now changed to:

“Most significantly enriched motif identified in conserved m⁶A peaks across cellular poly(A)+RNA in CHIKV-infected cells.”

6. In Figs 2d and 6g, the authors should indicate positions of the qPCR primers or the qPCR product relative to the m⁶A peaks.

We have now annotated the 11 qPCR products relative to the m⁶A peaks for both figures (and the amplicon that spans the viral m⁶A peaks) and included the positive and negative amplicons for the *SLC39A14* RNA. Note that all qPCR negative control primers used in m⁶A-IP-qPCR (with RNA fragmented down to 100 nt) are positioned at least 300 nt away from the m⁶A peaks.

New figure 2b and 2c.

New figure 6f.

7. In line 114, the sentence describing the ORFs of the gRNA is confusing. Are structural proteins expressed from both the sgRNA and gRNA second ORF, or only the sgRNA?

We apologize for the generated confusion and have modified the text (page 5).

“The CHIKV (+) RNA genome (gRNA) consists of a 5' capped and 3' poly(A)-tailed single-stranded RNA that contains two open reading frames (ORFs). The first ORF encodes four non-structural proteins required for RNA replication. The second ORF is expressed from a 5' capped and 3' poly(A)-tailed subgenomic RNA (sgRNA) transcribed during infection and encodes five structural proteins found within the virion.”

8. In Supp Fig 4, the authors should include a graph showing NanoConsensus scores of the Adenovirus transcript from the METTL3 KO cells.

We thank the reviewer for his/her suggestion. However, we would like to note that NanoConsensus performs comparative analysis of samples in a pairwise manner and detects differentially modified sites between them. As a result, it only produces one set of tracks per each pairwise comparison, and not one track per sample. The tracks provided in Figure S3 are the ones generated by the algorithm when comparing Ad5 wild-type and METTL3 knockout samples.

9. In Supp Figs 5, 6, and 10, can the authors include quantification of nuclear:cytoplasmic signal in mock VS infected? It seems that there is more WTAP in the cytoplasm after infection with DENV (which has been previously reported with other types of viruses) but it is not clear from the images.

We have now performed the quantification requested for CHIKV in U2OS cells (Figure S6) and for DENV in Huh7 cells (Figure S10). We were unable to quantify nuclear:cytoplasmic in HEK293T cells (Figure S5) as these cells hardly have cytoplasm. Analyses in Huh7 and U2OS cells showed that WTAP signal in the cytoplasm does not increase under infection conditions. These analyses have been incorporated as Figure S11 and incorporated in the results section (page 11).

Supplementary Fig. 11. Quantification of the nuclear and cytoplasmic fluorescent signal for WTAP, in non-infected versus infected cells. a) Confocal micrographs of mock- or CHIKV (18 h p.i., MOI of 4)-infected U2OS cells were used for quantification. n = 4 for non-infected and 7 for infected. b) Confocal micrographs of mock- or DENV (48 h p.i., MOI of 3)-infected Huh7 cells were used for quantification. n = 11 for non-infected and 13 for infected. Graphs show the mean (of all mean gray values) with error bars representing standard error of the mean (SEM). All statistical analyses were performed using a two-tailed t-test. n.s. = not significant.

REVIEWERS' COMMENTS

Reviewer #2 (Remarks to the Author):

The three sets of reviews were in good alignment and for the most part, asked for clarifications or provided suggestions aimed at making the studies more accessible to readers. The authors have responded well providing additional information, including some additional analyses and in one case a reanalysis of published data sets. My concerns have been addressed. Again I think this is well well-crafted study and a very important addition to the m6A literature.

Reviewer #3 (Remarks to the Author):

I think the manuscript, which was already very good, has been improved with the few suggestions that were raised in the initial round of review.

Two comments that can be addressed by textual edits:

1. In the new text that was added to the discussion, the authors discuss some of the potential problems with m6A-Seq and antibody-based mapping methods. I recognize that these authors have pioneered antibody-independent methods, but that doesn't necessarily mean that they should criticize antibody-dependent methods with a broad brush. Antibody-based mapping methods have been very essential and have been highly accurate in terms of identifying a vast number of m6A sites and the core principles of m6A localization. The problem arises when people use the antibody-based methods improperly. In the case of mapping m6A in mRNAs, it's relatively easy to find peaks that are anomalously high because there are many other similarly abundant transcripts that can be used as comparisons and which lack these large peaks. However, when the viruses are used, the abundance of the virus is so high and there is a lack of other comparable-in-abundance transcripts that can be used as internal controls. Thus it is difficult to set up criteria for m6A peaks with viral RNAs, and the frankly the authors of the original studies used a bad peak calling approach that did not use valid controls like modification-free transcriptomes that can be synthesized by in vitro transcription, or, spiked in m6A-containing standards. It's easy to make improper conclusions when the mapping assays are performed incorrectly. However, incorrectly performed experiments should not be used to broadly criticize the method, especially when many groups use the methods properly and achieve largely valid mapping results. I would encourage the authors to have a slightly more balanced and nuanced discussion of antibody-based methods, and point out that the methods need to be performed properly with appropriate controls in order to ensure that the map sites are in fact real, also that many antibody-based mapping studies are performed correctly, but frankly they were missing from the viral m6A mapping studies that are being challenged here.

2. YTHDF1 was tested in the present manuscript, but it is worth mentioning that other readers were not tested, including YTHDF2 and YTHDF3, which have been shown by some groups to be functionally redundant. I would suggest a comment when YTHDF1 is initially tested like "other m6A readers, such as YTHDF2 and YTHDF3, which some have shown to be functionally redundant with YTHDF1, were not tested here since previous studies examined YTHDF1" or something similar.

Reply to REVIEWER #3

1. In the new text that was added to the discussion, the authors discuss some of the potential problems with m6A-Seq and antibody-based mapping methods. I recognize that these authors have pioneered antibody-independent methods, but that doesn't necessarily mean that they should criticize antibody-dependent methods with a broad brush. Antibody-based mapping methods have been very essential and have been highly accurate in terms of identifying a vast number of m6A sites and the core principles of m6A localization. The problem arises when people use the antibody-based methods improperly. In the case of mapping m6A in mRNAs, it's relatively easy to find peaks that are anomalously high because there are many other similarly abundant transcripts that can be used as comparisons and which lack these large peaks. However, when the viruses are used, the abundance of the virus is so high and there is a lack of other comparable-in-abundance transcripts that can be used as internal controls. Thus it is difficult to set up criteria for m6A peaks with viral RNAs, and frankly the authors of the original studies used a bad peak calling approach that did not use valid controls like modification-free transcriptomes that can be synthesized by in vitro transcription, or, spiked in m6A-containing standards. It's easy to make improper conclusions when the mapping assays are performed incorrectly. However, incorrectly performed experiments should not be used to broadly criticize the method, especially when many groups use the methods properly and achieve largely valid mapping results. I would encourage the authors to have a slightly more balanced and nuanced discussion of antibody-based methods, and point out that the methods need to be performed properly with appropriate controls in order to ensure that the map sites are in fact real, also that many antibody-based mapping studies are performed correctly, but frankly they were missing from the viral m6A mapping studies that are being challenged here.

Reply:

We believe that these points are already addressed in the current discussion and would prefer not to change the text. For example:

“ These drawbacks are diminished when highly m⁶A-modified transcripts are scrutinised, as in the case of nuclear-replicating viruses, such as Kaposi's sarcoma-associated herpesvirus, in which highly reproducible m⁶A peaks have been reported and validated by multiple independent groups¹⁻³

or

” Our findings highlight the critical importance of employing orthogonal validation methods and standardised controls when assessing the presence of m6A modification in the genomes of exclusively cytoplasmic viruses. To ensure the robustness of m6A detection, different strategies should be implemented, including antibody-independent techniques, along with the establishment of consistent positive and negative controls. Ideally, this should include described m6A modifications within host RNAs and viral IVT RNAs, respectively”

Our experience has shown that as for any other method, orthogonal validations are essential in m⁶A detection.

2. YTHDF1 was tested in the present manuscript, but it is worth mentioning that other readers were not tested, including YTHDF2 and YTHDF3, which have been shown by some groups to be functionally redundant. I would suggest a comment when YTHDF1 is initially tested like "other m⁶A readers, such as YTHDF2 and YTHDF3, which some have shown to be functionally redundant with YTHDF1, were not tested here since previous studies examined YTHDF1" or something similar.

Reply:

We have included the following sentence to address this comment (highlighted in yellow in the final manuscript):

Other m⁶A readers, such as YTHDF2 or YTHDF3, were not assessed in this study, as it was recently shown that these proteins are functionally redundant⁴ and in general the knockdown of these proteins results in a similar outcome of viral infection in multiple viruses⁵. Therefore, we only depleted YTHDF1 as a representative m⁶A reader.

1. Baquero-Perez, B. *et al.* The Tudor SND1 protein is an m⁶A RNA reader essential for replication of Kaposi's sarcoma-associated herpesvirus. *eLife* **8**, e47261 (2019).
2. Hesser, C.R., Karjolich, J., Dominissini, D., He, C. & Glaunsinger, B.A. N⁶-methyladenosine modification and the YTHDF2 reader protein play cell type specific roles in lytic viral gene expression during Kaposi's sarcoma-associated herpesvirus infection. *PLOS Pathogens* **14**, e1006995 (2018).
3. Tan, B. *et al.* Viral and cellular N⁶-methyladenosine and N⁶,2'-O-dimethyladenosine epitranscriptomes in the KSHV life cycle. *Nature Microbiology* **3**, 108-120 (2018).
4. Zaccara, S. & Jaffrey, S.R. A Unified Model for the Function of YTHDF Proteins in Regulating m⁶A-Modified mRNA. *Cell* **181**, 1582-1595.e1518 (2020).
5. Baquero-Perez, B., Geers, D. & Díez, J. From A to m⁶A: The Emerging Viral Epitranscriptome. *Viruses* **13**, 1049 (2021).